# ComPO: Preference Alignment via Comparison Oracles

**Peter Chen**◇    **Xi Chen**†    **Wotao Yin**‡    **Tianyi Lin**◇

Columbia University◇

Stern School of Business, New York University†

DAMO Academy, Alibaba Group US‡

{lc3826, tl3335}@columbia.edu, xc13@stern.nyu.edu

wotao.yin@alibaba-inc.com

## Abstract

Direct alignment methods are increasingly used for aligning large language models (LLMs) with human preferences. However, these methods suffer from the issues of *likelihood displacement*, which can be driven by noisy preference pairs that induce similar likelihood for preferred and dispreferred responses. The contributions of this paper are two-fold. First, we propose a preference alignment method based on zeroth-order, comparison-based optimization via comparison oracles and provide convergence guarantees for its basic mechanism. Second, we improve our method using some heuristics and conduct the experiments to demonstrate the flexibility and compatibility of practical mechanisms in improving the performance of LLMs using noisy preference pairs. Evaluations are conducted across multiple base and instruction-tuned models (Mistral-7B, Llama-3-8B and Gemma-2-9B) with benchmarks (AlpacaEval 2, MT-Bench and Arena-Hard)[1]. Experimental results show the effectiveness of our method as an alternative to addressing the limitations of existing methods, not only *likelihood displacement* but *verbosity*. A highlight of our work is that we evidence the importance of designing specialized methods for preference pairs with distinct likelihood margin, which complements the recent findings in Razin et al. [73].

## 1    Introduction

Generative AI is breaking down barriers to intelligence, empowering domain experts across academia, industrial sectors, and governments to develop and manage AI systems more effectively. At the heart of this revolution are large language models (LLMs), which are transforming data organization, retrieval, and analysis [9, 20, 85, 1, 10]. These models are trained on vast, diverse data and must be carefully aligned with human preferences to ensure they generate helpful and harmless content [7]. A prominent alignment method is *reinforcement learning from human feedback* (RLHF) [21, 81], which first fits a reward model based on human preference pairs and then uses RL to train a policy to maximize this trained reward. Despite RLHF's success [108, 63, 85, 1], it involves a complex and computationally expensive multi-stage procedure. This motivates the development of direct alignment methods, such as direct preference optimization (DPO) [70] and its variants [6, 30, 66, 93, 83, 60, 16, 103], which directly optimize the LLM using the human preference pairs, avoiding the need for separately training a reward model.

The applications of direct alignment methods have gained momentum due to their simplicity and training stability. However, these methods suffer from one critical issue: likelihood displacement. *Likelihood displacement* refers the counterintuitive situation where the training process, designed to increase

---

[1]Models and code: `huggingface.co/ComparisonPO`  `github.com/PeterLauLukChen/ComparisonPO`

39th Conference on Neural Information Processing Systems (NeurIPS 2025).

the likelihood of preferred responses relative to dispreferred ones, actually reduces the absolute probability of the preferred responses, leading to "unintentional unalignment" [64, 82, 72, 65, 57, 96, 73]. For example, training a model to prefer NO over NEVER sharply increase the likelihood of YES. Practically, this issue has negative impacts on LLM performance: it unintentionally shifts probability mass to harmful responses. For example, if the prompt is to outline the steps for a terrorist organization to infiltrate a government agency, Gemma-2B-it initially generates two refusal responses. After DPO training, it complies with unsafe prompts due to likelihood displacement shifting probability mass from the preferred refusal responses to harmful responses (see [73, Table 18]). There is another issue – *verbosity* – which refers to the tendency of models fine-tuned with RLHF [78, 44] and direct alignment methods [66, 4, 71] to generate longer responses, often without corresponding improvement in quality, and which causes low efficiency and higher consumption of hardware resources.

Recent work has suggested that likelihood displacement might arise from preference pairs that induce similar preferred and displeased responses [64, 73] (which we refer to them as *noisy* preference pairs in this paper). To mitigate this issue, researchers have tried to add different regularization [64, 72]. Recently, Razin et al. [73] proposed to measure the similarity between preferred and dispreferred responses using the centered hidden embedding similarity (CHES) score and empirically showed that filtering out the preference pairs with small CHES score is more effective for mitigating the likelihood displacement compared to adding the supervised fine-tuning (SFT) regularization.

While DPO provides a computationally convenient framework by maximizing certain log-likelihood margin between preferred and dispreferred responses, this objective function seems to act as a *proxy* or *surrogate* for the true, complex goal of alignment. This proxy works well enough when preference pairs clearly delineate better responses. However, when faced with noisy pairs — *where the preference signal is weak or ambiguous, relative to the embeddings of the two responses* — optimizing the specific DPO objective function can result in adverse effects such as likelihood displacement, as this objective function itself does not accurately reflect the desired alignment improvement. Formally and explicitly defining alignment as a single, optimizable mathematical objective function is exceptionally challenging. Instead of pursuing such an explicit objective, we shall recognize that preference pairs inherently represent comparative judgments based on this underlying, albeit latent, alignment goal. Inspired by comparison-oracle-based optimization techniques, which navigate in search spaces using only comparison outcome information (that is, "is solution A better than solution B?"), we leverage preference pairs in a similar manner. We treat the preference pairs as the oracle outputs based on the hidden alignment objective. This allows us to use them to guide model parameter updates directly, thus avoiding a commitment to an explicit proxy objective.

**Contribution.** In this paper, we propose a preference alignment method that directly leverages comparative preference pairs by employing comparison oracles and effectively utilize the signals present even in noisy preference pairs. Our contributions can be summarized as follows:

1. We identify that likelihood displacement issue is exacerbated by the ineffective handling of noisy preference pairs in existing methods. We propose to mitigate this issue by developing a method based on a specialized comparison oracle to extract useful information from these pairs. We also provide a convergence guarantee for the basic scheme of our method under non-convex, smooth settings.

2. To ensure computational efficiency for large-scale model fine-tuning, we enhance our method with several techniques, including integration with DPO to handle clean and noisy preference data separately and approximating expensive steps in standard comparison-oracle-based optimization by efficiently restricting and clipping normalized gradients.

3. We conduct extensive experiments demonstrating the flexibility and effectiveness of our practical approach in improving LLM performance, particularly leveraging both clean and noisy preference data. Evaluations are undertaken across base and instruction-tuned models (Mistral-7B, Llama-3-8B, and Gemma-2-9B) using benchmarks (AlpacaEval 2, MT-Bench, and Arena-Hard). Experimental results validate our approach's effectiveness, which addresses limitations in current direct alignment techniques.

Recent works [32, 27, 66, 4, 93, 60, 65] have shown that *verbosity* can be mitigated by incorporating appropriate regularization into the objective, suggesting that modifying the objective could better capture alignment goals. Although our method is not specifically designed for addressing verbosity issue, it consistently improve length-controlled win rate (LC), indicating that our method helps reduce verbosity by possibly optimizing a more robust and alignment-faithful objective.

**Related works.** Our work is mainly connected to the literature on direct preference alignment methods and optimization techniques utilizing comparison oracles. Due to space limitations, we defer our comments on other relevant topics to Appendix B. Direct preference alignment methods (such as DPO [70]) are simple and stable offline alternatives to RLHF. Various DPO variants with other objectives were proposed, including ranking ones beyond pairwise preference data [25, 95, 79, 15, 56] and simple ones that do not rely on a reference model [38, 60]. It is well known that DPO suffers from the issues of verbosity [66, 4, 71] and likelihood displacement [64, 82, 72, 65, 57, 96], which can be interpreted from a unified perspective of data curation [66, 73]. Our work continues along this perspective by arguing that these issues can be mitigated by using the information contained in the noisy preference pairs that induce similar likelihood for preferred and dispreferred responses.

In optimization literature, the first algorithm based on comparison oracles is a variant of the coordinate descent method [41, 59], and two representative methods that consider using comparison oracles to approximate the gradient are [12, 18]. The major drawback in these works is that the objective function is assumed to be convex or strongly convex, which is unrealistic in preference alignment applications. There are other works that investigate the value of comparison oracles in the context of online bandit optimization [98, 47, 24], Bayesian optimization [5, 53] and RLHF [84, 100]. Our work extends [12] to preference alignment through *nontrivial* modifications. First, we define the comparison oracle based on response likelihood: $\theta_1$ is better than $\theta_2$ if $\theta_1$ achieves a higher likelihood for preferred responses and a lower likelihood for dispreferred responses. Second, we prove a convergence rate guarantee beyond the convex settings. Finally, instead of explicitly imposing a sparsity constraint when estimating normalized gradients, which leads to an expensive computational sub-step, we reduce the computation by approximating the sub-step by clipping an approximated normalized gradient, which is efficient for fine-tuning LLMs. The key difference between our work and recent works [84, 100] is the specific design of our comparison oracle, which is constructed to extract meaningful directional information from noisy preference pairs prevalent in alignment datasets, thereby mitigating verbosity and likelihood displacement.

## 2 Preliminaries and Technical Background

We provide an overview of the setup for direct preference alignment in this paper, and the definition for comparison oracles and the subroutine for estimating gradients using comparison oracles that are important to designing the basic scheme of our method.

### 2.1 Direct preference alignment

Modern LLMs are designed based on the Transformer architecture [86] and follow user prompts $\mathbf{x} \in \mathcal{V}^\star$ to generate a response $\mathbf{y} \in \mathcal{V}^\star$, where $\mathcal{V}$ is a vocabulary of tokens. We consider an LLM as a policy $\pi_\theta(\mathbf{y}|\mathbf{x})$ which corresponds to probabilities to $\mathbf{y}$ given $\mathbf{x}$. For assigning probabilities to each token of $\mathbf{y}$, the policy $\pi_\theta$ operates in an auto-regressive manner as follows,

$$\pi_\theta(\mathbf{y}|\mathbf{x}) = \Pi_{k=1}^{|\mathbf{y}|} \pi_\theta(\mathbf{y}_k|\mathbf{x}, \mathbf{y}_{<k}),$$

where $\theta$ stands for the model's parameter (e.g., the parameters of the Transformer architecture) and $\mathbf{y}_{<k}$ denotes the first $k-1$ tokens of $\mathbf{y}$. However, the generated responses might not be helpful, safe or reliable, which necessities the process of further aligning the LLMs with human preference.

We consider the direct preference learning pipeline which relies on pairwise preference data. Indeed, we assume the access to a preference dataset $\mathcal{D}$ containing samples $(\mathbf{x}, \mathbf{y}^+, \mathbf{y}^-)$, where $\mathbf{x}$ is a prompt and $(\mathbf{y}^+, \mathbf{y}^-)$ is a pair of preferred and dispreferred responses to $\mathbf{x}$. This pipeline includes an initial supervised fine-tuning (SFT) phase where the model is fine-tuned using the cross-entropy loss and high-quality data for specific downstream tasks. The SFT data can be either independent of $\mathcal{D}$ [85] or consists of prompts and preferred responses from $\mathcal{D}$ [70].

Direct alignment methods (e.g., DPO [70]) optimize the policy $\pi_\theta$ over the preference dataset $\mathcal{D}$ without learning a reward model as in RLHF [108, 81]. This is typically done by minimizing a loss of the following form:

$$\mathcal{L}_{\mathrm{DPO}}(\theta) = -\mathbb{E}_{(\mathbf{x}, \mathbf{y}^+, \mathbf{y}^-) \sim \mathcal{D}} \left[ \log \sigma \left( \beta \log \frac{\pi_\theta(\mathbf{y}^+|\mathbf{x})}{\pi_{\mathrm{ref}}(\mathbf{y}^+|\mathbf{x})} - \beta \log \frac{\pi_\theta(\mathbf{y}^-|\mathbf{x})}{\pi_{\mathrm{ref}}(\mathbf{y}^-|\mathbf{x})} \right) \right], \tag{1}$$

where $\pi_{\mathrm{ref}}$ is the model after SFT, $\beta$ is a regularization parameter, and $\sigma : \mathbb{R} \mapsto [0, 1]$ is the sigmoid function. However, the function $\mathcal{L}_{\mathrm{DPO}}$ relies on the log-likelihood margin between $\mathbf{y}^+$ and $\mathbf{y}^-$ such

that DPO maximizes the likelihood margin between $\mathbf{y}^+$ and $\mathbf{y}^-$ rather than maximizing the likelihood for $\mathbf{y}^+$ and minimizing the likelihood for $\mathbf{y}^-$. The likelihood of $\mathbf{y}^+$ might decrease during training and the probability mass is shifted from $\mathbf{y}^+$ to responses with an opposite meaning [64, 73]. One of possible reasons is that the above objective function is not suitable for extracting information from noisy preference pairs that induce similar preferred and dispreferred responses.

Empirically, [73] has shown that filtering out similar preference pairs makes DPO more effective. However, the noisy preference pairs might contain rich information that can improve the performance of LLMs. Extracting such information is challenging since it is difficult to explicitly write down an objective function that maximizes the likelihood for $\mathbf{y}^+$ and minimizes the likelihood for $\mathbf{y}^-$, and the only thing that we know is that its function value is smaller for a better policy which exhibits a higher likelihood for $\mathbf{y}^+$ and a lower likelihood for $\mathbf{y}^-$. This motivates us to design a new alignment method by directly leveraging the comparison signal in pairwise preference data $(\mathbf{x}, \mathbf{y}^+, \mathbf{y}^-)$ from $\mathcal{D}$.

## 2.2 Comparison oracles and zeroth-order methods

To contextualize our proposed method for aligning LLMs with human preferences, we review the definitions for comparison oracles and explain how one leverages the comparison oracles to develop the zeroth-order methods in the literature.

Given that $f : \mathbb{R}^d \to \mathbb{R}$ is a function where neither its function value nor its gradient is accessible, we define a pairwise comparison oracle $\mathcal{C}_f$ in its simplest form as follows,

**Definition 2.1** *We call $\mathcal{C}_f(\theta, \theta') : \mathbb{R}^d \times \mathbb{R}^d \to \{+1, -1\}$ a comparison oracle for function $f$ if*

$$\mathcal{C}_f(\theta, \theta') = \begin{cases} -1, & \text{if } f(\theta') < f(\theta), \\ +1, & \text{otherwise.} \end{cases}$$

*In other words, when separately queried with $\theta$ and $\theta'$, the oracle $\mathcal{C}_f$ returns $\text{sign}(f(\theta') - f(\theta))$.*

The key idea of designing the subroutine in [12] for estimating gradients using comparison oracles comes from 1-bit compressed sensing [8]. The goal is to recover a signal $\mathbf{g} \in \mathbb{R}^d$ from the quantized measurements $y_i = \text{sign}(\mathbf{z}_i^\top \mathbf{g})$ where $\mathbf{z}_i$ is a random perturbation vector drawn from any rationally invariant distribution. The theoretical guarantee on the required number of perturbations to obtain an approximate signal was established in [69] and extended in [12]. Notably, we have

$$\mathcal{C}_f(\theta, \theta + r\mathbf{z}_i) \approx \text{sign}(f(\theta + r\mathbf{z}_i) - f(\theta)) \approx \text{sign}(\mathbf{z}_i^\top \nabla f(\theta)),$$

where $r > 0$ is a parameter that controls the magnitude of perturbation. As such, the comparison oracle returns $y_i = \mathcal{C}_f(\theta, \theta + r\mathbf{z}_i)$ which serves as an approximate 1-bit measurement of $\nabla f(\theta)$.

Another crucial issue is that the zeroth-order comparison-based methods suffers from the dimension-dependent iteration complexity bound [41]. This makes sense since the comparison oracles are even weaker than the function value oracles. Such issue can be mitigated through exploiting sparse gradient structure [89, 35, 19, 12, 13]. Indeed, the high-dimensional function $f$ has sparse gradients satisfying that $\|\nabla f(\theta)\|_1 \leq \sqrt{s}\|\nabla f(\theta)\|$ for all $\theta \in \mathbb{R}^d$ and some $s \ll d$.

The above discussions give the subroutine for estimating sparse gradients using comparison oracles. We generate $m$ i.i.d. perturbation vectors from a uniform distribution (i.e., $\{\mathbf{z}_i\}_{1 \leq i \leq m}$), compute $y_i = \mathcal{C}_f(\theta, \theta + r\mathbf{z}_i)$ for all $i$, and solve the optimization problem in the following form of

$$\hat{\mathbf{g}} = \underset{\|\mathbf{g}\|_1 \leq \sqrt{s}, \|\mathbf{g}\| \leq 1}{\text{argmax}} \sum_{i=1}^{m} y_i \mathbf{z}_i^\top \mathbf{g}, \tag{2}$$

where the constraints $\|\mathbf{g}\|_1 \leq \sqrt{s}$ and $\|\mathbf{g}\|_2 \leq 1$ ensure that $\hat{\mathbf{g}}$ is sparse and normalized.

## 3 Main Results

We study how to use the information contained in noisy preference pairs that induce similar likelihood for preferred and dispreferred responses and achieve this goal by developing a zeroth-order preference alignment method based on comparison oracles. We provide the convergence guarantee for the basic scheme and improve it to the practical scheme using some heuristics.

---

**Algorithm 1** Comparison-Based Preference Alignment (Basic Scheme)

---

1: **Input**: initial parameter $\theta_1 \in \mathbb{R}^d$, stepsize $\eta > 0$, sparsity ratio $s \ll d$, sampling radius $r > 0$, querying number $m \geq 1$, and iteration number $T \geq 1$.
2: **for** $t = 1, 2, \ldots, T$ **do**
3:     Draw $m$ i.i.d. samples uniformly from a unit sphere in $\mathbb{R}^d$, i.e., $\{\mathbf{z}_i\}_{1 \leq i \leq m}$.
4:     Compute $y_i = \mathcal{C}_\pi(\theta_t, \theta_t + r\mathbf{z}_i)$ for $i = 1, 2, \ldots, m$.
5:     Compute $\hat{\mathbf{g}}_t = \mathrm{argmax}_{\|\mathbf{g}\|_1 \leq \sqrt{s}, \|\mathbf{g}\| \leq 1} \sum_{i=1}^m y_i \mathbf{z}_i^\top \mathbf{g}$.
6:     Compute $\theta_{t+1} = \theta_t - \eta\hat{\mathbf{g}}_t$.

---

### 3.1 Basic scheme with convergence guarantee

We adapt the comparison oracles to the setup for direct preference alignment. Instead of optimizing the DPO objective function in Eq. (1), we assume that there exists an *appropriate* objective function $f(\theta)$ that better aligns the LLMs with human preferences and optimize it. This function is complicated such that the function value oracles will not be accessible. However, it intuitively makes sense that $f(\theta') < f(\theta)$ if and only if $\pi_{\theta'}$ exhibits a higher likelihood for $\mathbf{y}^+$ and a lower likelihood for $\mathbf{y}^-$ than $\pi_\theta$ given any pairwise preference data $(\mathbf{x}, \mathbf{y}^+, \mathbf{y}^-)$. Based on these insights, we have

**Definition 3.1** *We say $\mathcal{C}_\pi(\theta, \theta') : \mathbb{R}^d \times \mathbb{R}^d \rightarrow \{+1, -1\}$ a preference comparison oracle for the model $\pi_\theta$ and a pair of preference data $(\mathbf{x}, \mathbf{y}^+, \mathbf{y}^-)$ from the offline dataset $\mathcal{D}$ if*

$$\mathcal{C}_\pi(\theta, \theta') = \begin{cases} -1, & \text{if } \pi_{\theta'}(\mathbf{y}^+|\mathbf{x}) > \pi_\theta(\mathbf{y}^+|\mathbf{x}) \text{ and } \pi_{\theta'}(\mathbf{y}^-|\mathbf{x}) < \pi_\theta(\mathbf{y}^-|\mathbf{x}) \text{ for } (\mathbf{x}, \mathbf{y}^+, \mathbf{y}^-) \in \mathcal{D}, \\ +1, & \text{otherwise.} \end{cases}$$

It is worth remarking that the oracle $\mathcal{C}_\pi(\theta, \theta')$ provides a preference comparison between parameters $\theta$ and $\theta'$ based on the model $\pi_\theta$ and the offline preference data from $\mathcal{D}$. Indeed, $\mathcal{C}_\pi(\theta, \theta') = -1$ indicates that $\pi_{\theta'}$ is a better model compared to $\pi_{\theta_1}$ for the offline preference dataset $\mathcal{D}$. By leveraging these comparative assessments, our goal is to find the parameter $\theta^\star$ that minimizes the function $f(\theta)$ which can be *nonconvex* in general.

We present the basic scheme of our method in Algorithm 1, which can be interpreted as a variant of the method [12]. It combines 1-bit gradient estimator from Eq. (2) with preference comparison oracles. However, the underlying objective function is assumed to be convex in [12], which is unrealistic in aligning LLMs with human preference. In what follows, we provide the convergence guarantee for Algorithm 1 given that there exists a smooth yet *nonconvex* function $f$ which can be compatible with the preference comparison oracle and has sparse gradients.

**Theorem 3.1** *Suppose that there exists a smooth function $f$ satisfying (i) $f(\theta') < f(\theta)$ if and only if $\pi_{\theta'}(\mathbf{y}^+|\mathbf{x}) > \pi_\theta(\mathbf{y}^+|\mathbf{x})$ and $\pi_{\theta'}(\mathbf{y}^-|\mathbf{x}) < \pi_\theta(\mathbf{y}^-|\mathbf{x})$ for $\forall(\mathbf{x}, \mathbf{y}^+, \mathbf{y}^-) \in \mathcal{D}$ and (ii) $\|\nabla f(\theta)\|_1 \leq \sqrt{s}\|\nabla f(\theta)\|$. For any $\epsilon, \Lambda \in (0, 1)$, there exists some $T > 0$ such that the output of Algorithm 1 with $\eta = \sqrt{\frac{2\Delta}{\ell T}}$, $r = \frac{\epsilon}{40\ell\sqrt{d}}$ and $m = c_m(s \log(\frac{2d}{s}) + \log(\frac{\ell\Delta}{\Lambda\epsilon^2}))$ (where $c_m > 0$ is a constant) satisfies that $\mathbb{P}(\min_{1 \leq t \leq T} \|\nabla f(\theta_t)\|\| < \epsilon) > 1 - \Lambda$ and the total number of calls of the preference comparison oracles is bounded by*

$$O\left(\frac{\ell\Delta}{\epsilon^2}\left(s \log\left(\frac{2d}{s}\right) + \log\left(\frac{\ell\Delta}{\Lambda\epsilon^2}\right)\right)\right),$$

*where $\ell > 0$ is the smoothness parameter of $f$ (i.e., $\|\nabla f(\theta) - \nabla f(\theta')\| \leq \ell\|\theta - \theta'\|$) and $\Delta > 0$ is an upper bound for the initial objective function gap, $f(\theta_1) - \inf_\theta f(\theta) > 0$.*

**Remark 3.2** *It is worth mentioning that we derive $\mathbb{P}(\min_{1 \leq t \leq T} \|\nabla f(\theta_t)\|\| < \epsilon) > 1 - \Lambda$ in the analysis (see also [62, 52]) but finding the best solution from $\{\theta_1, \ldots, \theta_T\}$ is intractable since $\|\nabla f(\theta_t)\|$ can not be estimated in practice. Nonetheless, this best-iterate guarantee has been also used in other recent works that leverage comparison oracles to RLHF [84] and can be viewed as a theoretical benchmark. In addition, the sample complexity bound is independent of $d \geq 1$ up to a logarithmic factor thanks to gradient sparsity structure.*

---

**Algorithm 2** Comparison-Based Preference Alignment (Practical Scheme)

---

1: **Input**: initial parameter $\theta_1 = [\bar{\theta}_1; \theta_t^\mathrm{o}] \in \mathbb{R}^d$, scaling for stepsize $\gamma > 0$, sampling radius $r > 0$, querying number $m \geq 1$, clipping thresholds $\lambda_g, \lambda > 0$, and iteration number $T \geq 1$.
2: **for** $t = 1, 2, \ldots, T$ **do**
3:     Draw $m$ i.i.d. samples uniformly from a unit sphere in $\mathbb{R}^{d^\mathrm{o}}$, i.e., $\{\mathbf{z}_i\}_{1 \leq i \leq m}$.
4:     Compute $y_i = \mathcal{C}_\pi([\bar{\theta}_1; \theta_t^\mathrm{o}], [\bar{\theta}_1; \theta_t^\mathrm{o} + r\mathbf{z}_i])$ for $i = 1, 2, \ldots, m$.
5:     Compute $\hat{\mathbf{g}}_t^\mathrm{o} = \frac{\sum_{i=1}^m y_i \mathbf{z}_i}{\| \sum_{i=1}^m y_i \mathbf{z}_i \|}$ and clip $\hat{\mathbf{g}}_t^\mathrm{o}$ by zeroing out the entries whose magnitude is less than $\lambda_g$.
6:     Compute $\theta_{t+1}^\mathrm{o} = \theta_t^\mathrm{o} - \frac{\gamma |\{i : y_i = -1\}|}{m} \hat{\mathbf{g}}_t^\mathrm{o}$ if $\frac{|\{i : y_i = -1\}|}{m} > \lambda$ and $\theta_{t+1}^\mathrm{o} = \theta_t^\mathrm{o}$ otherwise.

---

## 3.2   Practical scheme

It is clear that the basic scheme in Algorithm 1 is impractical primarily because the dimension $d$ is at a billion level. The steps, including weight perturbation and 1-bit gradient estimation, are intractable due to their memory and computation demands. We also hope to incorporate gradient clipping [67, 33, 61, 68, 99] to stabilize our method in practice.

**Perturbations on output layer weights.**   Algorithm 1 consists of performing $m$ perturbations on all model parameters, which is unacceptable due to the high computational and memory costs. To reduce costs, we restrict the perturbations to only the *output layer weights* in $\theta$ and draw $m$ i.i.d. samples uniformly from a unit sphere in $\mathbb{R}^{d^\mathrm{o}}$ where $d^\mathrm{o}$ is the number of output layer weights.

**Low-cost approximation to normalized sparse gradient estimation.**   The $\ell_1$-norm constrained normalized gradient estimation problem (2) becomes computationally intractable when $d$ reaches billions. We propose a practical approximation by first relaxing the $\ell_1$-norm constraint and then applying clipping. Specifically, we first compute the normalized gradient of the output layer:

$$\hat{\mathbf{g}}^\mathrm{o} = \frac{\sum_{i=1}^m y_i \mathbf{z}_i}{\| \sum_{i=1}^m y_i \mathbf{z}_i \|}.$$

In the clipping step, we zero out the entries of $\hat{\mathbf{g}}^\mathrm{o}$ falling below a certain threshold $\lambda_g$. The restriction–normalization–clipping operations yield an approximate sparse solution to (2) for the output layer weights. In addition, we adjust the stepsize based on $\{y_i\}_{1 \leq i \leq m}$. Intuitively, if the size of $\{i : y_i = -1\}$ is larger, it is more likely that $\hat{\mathbf{g}}$ obtained from $\{(\mathbf{z}_i, y_i)\}_{1 \leq i \leq m}$ leads to much progress for minimizing the function $f(\theta)$. This motivates us to use a larger stepsize. That being said, the stepsize is proportional to $\frac{|\{i : y_i = -1\}|}{m}$ if $\frac{|\{i : y_i = -1\}|}{m}$ is relatively large. However, if the size of $\{i : y_i = -1\}$ falls below a threshold $\lambda$, the estimator $\hat{\mathbf{g}}$ lacks sufficient information, so we should skip it.

We summarize the practical scheme of our method in Algorithm 2. We can view this algorithm as a micro-finetuning approach designed for noisy preference pairs, which serves as a practical addition to existing direct preference alignment methods, such as DPO [70] and SimPO [60].

**Final method.**   By combining the above algorithm with existing methods, we propose a unified preference alignment framework that leverages both clean and noisy pairwise preference data. It consists of three steps: we first use a reference model to divide the dataset into two subsets: clean and noisy. Then, we apply DPO on the clean preference pairs to obtain an initial policy; we label it $\mathrm{DPO}_{\mathrm{clean}}$ to differentiate it from applying DPO to all the data. In the third step, starting from the initial policy of $\mathrm{DPO}_{\mathrm{clean}}$, we apply Algorithm 2 to only noisy preference pairs to obtain a final policy. We label all the three steps as $\mathrm{DPO}_{\mathrm{clean}} + \mathrm{ComPO}$.

Specifically, we say one pairwise preference data $(\mathbf{x}, \mathbf{y}^+, \mathbf{y}^-) \in \mathcal{D}$ is *noisy* if the log-likelihood for preferred and dispreferred responses is similar with respect to the reference model (after SFT). This can be formalized as follows,

$$\| \log \pi_{\mathrm{ref}}(\mathbf{y}^+|\mathbf{x}) - \log \pi_{\mathrm{ref}}(\mathbf{y}^-|\mathbf{x}) \| \leq \delta, \tag{3}$$

where $\delta > 0$ is a threshold. This curation of data is inspired by Razin et al. [73] who has shown that the issue of likelihood displacement can be mitigated by filtering out the pairwise preference data with small centered hidden embedding similarity (CHES) score. While a soft interpolation based on a reference-model-derived confidence score is a compelling idea, we argue that defining such noise

Table 1: Evaluation results on AlpacaEval 2, Arena-Hard, and MT-Bench under four model setups. LC and WR denote length-controlled win rate and win rate, respectively. Turn-1 and Turn-2 represent the scores to the answers from the first and follow-up questions in multi-turn dialogue. Here, we run 5 trials for $DPO_{clean}$+ComPO and present the best trail performance.

| Method | Mistral-Base-7B | | | | | | Mistral-Instruct-7B | | | | | |
| | AlpacaEval 2 | | Arena-Hard | MT-Bench | | | AlpacaEval 2 | | Arena-Hard | MT-Bench | | |
| | LC (%) | WR (%) | WR (%) | Turn-1 | Turn-2 | Avg. | LC (%) | WR (%) | WR (%) | Turn-1 | Turn-2 | Avg. |
| DPO | 9.71 | 6.27 | 2.9 | 6.20 | **5.38** | **5.79** | 24.14 | 16.71 | **14.4** | 6.28 | 5.42 | 5.86 |
| $DPO_{clean}$ | 9.41 | 6.52 | 3.0 | 6.18 | 5.22 | 5.70 | 23.89 | 16.15 | 14.2 | 6.11 | 5.34 | 5.73 |
| $DPO_{clean}$+ComPO | **11.66** | **6.55** | **3.2** | **6.22** | 5.32 | 5.77 | **26.17** | **18.32** | 10.5 | **7.78** | **7.63** | **7.69** |

| Method | Llama-3-Base-8B | | | | | | Llama-3-Instruct-8B | | | | | |
| | AlpacaEval 2 | | Arena-Hard | MT-Bench | | | AlpacaEval 2 | | Arena-Hard | MT-Bench | | |
| | LC (%) | WR (%) | WR (%) | Turn-1 | Turn-2 | Avg. | LC (%) | WR (%) | WR (%) | Turn-1 | Turn-2 | Avg. |
| DPO | 4.14 | 10.43 | **12.1** | 6.61 | 5.85 | 6.23 | 32.59 | 31.99 | 22.9 | 8.30 | 7.55 | 7.93 |
| $DPO_{clean}$ | 4.28 | 9.81 | 12.0 | **6.64** | 6.01 | 6.33 | 32.92 | 32.42 | 22.9 | 8.26 | 7.63 | 7.94 |
| $DPO_{clean}$+ComPO | **5.39** | **10.93** | **12.1** | 6.60 | **6.28** | **6.44** | **35.79** | **35.03** | **23.1** | **8.39** | **7.71** | **8.05** |

level of a dataset remains nontrivial without relying on thresholds such as $\delta$. Even with this threshold, it remains unclear how to determine a suitable confidence score and interpolation scheme.

Our experimental results (see Table 1) highlight that filtering out the pairwise preference data with small log-likelihood margin is not always helpful, which supports the superiority of the CHES score. Indeed, the CHES score is defined based on the model's embedding geometry and better captures the similarity between preferred and dispreferred responses, while our metric based on log-likelihood is weaker yet *easy to compute in practice*. Nonetheless, our experimental results demonstrate that, despite a *weaker* similarity metric, our new method can *effectively* extract the information from the noisy preference pairs to further improve the performance of LLMs by a large margin in terms of length-controlled win rate (LC) (the higher LC means less verbosity) and mitigate the issue of likelihood displacement (see Table 3).

## 4 Experiment

We investigate the effectiveness of ComPO on aligning the LLMs with noisy preference pairs as an alternative to DPO and its variants. The objectives of the experiments include: (1) a quantitative evaluation of length-controlled win rate (LC), win rate (WR) and likelihood displacement; (2) a quantitative evaluation of computational and memory efficiency. We split the samples using $\delta = 3$. For Mistral-7B models, we set $r = 0.0005$, $m = 1600$, $\lambda_g = 0.00022$ and $\lambda = 0.2$. For Llama-3-8B models and Gemma-2-it-9B model, we set $r = 0.00075$, $m = 1800$, $\lambda_g = 0.00008$ and $\lambda = 0.2$. For the detailed information on datasets, models, and evaluation benchmarks, we defer to Appendix D. All the experiments are implemented in Python 3.10 with PyTorch 2.5.1 with 30 NVIDIA A40 GPUs each with 46 GB memory, equipped with Ubuntu 22.04.5 LTS.

### 4.1 Augmenting DPO and SimPO

We show that our method can utilize the information contained in noisy preference pairs to improve DPO and SimPO, especially in terms of LC and with the strong model.

**DPO.** We separately train DPO and $DPO_{clean}$ + ComPO. Table 1 presents the performance in terms of both length-controlled win rate (LC) and win rate (WR). In addition to the best result achieved by $DPO_{clean}$ + ComPO, we present the *average performance* over 5 consecutive runs for all models and benchmarks in Table 8 (see Appendix B). The objective of doing this to evidence the robustness of our method in effectively leveraging noisy preference data pairs.

We have several interesting findings. First of all, $DPO_{clean}$ does not always outperform DPO but could be better when the model is strong (e.g., Llama-3-Instruct-8B). This is possibly because our metric based log-likelihood margin is too simple to capture the similarity between preferred and dispreferred response, demonstrating the superiority of the CHES score [73]. Nonetheless, our metric is easy to compute and our experimental results show that, despite weaker metric, our method utilizes the noisy

Table 2: Direct augmentation results on SimPO over different models and benchmarks.

| Model | Method | AlpacaEval 2 | | Arena-Hard | MT-Bench | | |
|---|---|---|---|---|---|---|---|
| | | LC (%) | WR (%) | WR (%) | Turn-1 | Turn-2 | Avg. |
| **Mistral-Instruct-7B** | SimPO | 40.22 | 41.18 | 20.8 | **7.94** | 7.31 | 7.62 |
| | SimPO + ComPO | **42.27** | **43.17** | **22.0** | 7.83 | **7.46** | **7.64** |
| **Llama-3-Instruct-8B** | SimPO | 48.71 | 43.66 | 36.3 | 7.91 | 7.42 | 7.66 |
| | SimPO + ComPO | **49.53** | **45.03** | **37.3** | **7.94** | **7.45** | **7.70** |
| **Gemma-2-it-9B** | SimPO | 60.36 | 55.59 | **61.1** | **9.07** | 8.47 | 8.77 |
| | SimPO + ComPO | **62.42** | **57.20** | **61.1** | 8.99 | **8.58** | **8.79** |

Table 3: The log-likelihood for preferred and dispreferred responses for 3 independent trials with $\gamma \in \{0.1, 1\}$ and the default values for all of other parameters. Each cell gives a pair of log-likelihood for preferred and dispreferred responses $(\log \pi_\theta(\mathbf{y}^+|\mathbf{x}), \log \pi_\theta(\mathbf{y}^-|\mathbf{x}))$ after one trail of training. The results are indeed different since the perturbations $\{\mathbf{z}_i\}_{1 \le i \le m}$ are different for Trial 1, Trial 2 and Trial 3. However, we find that the log-likelihood for preferred response increase and the log-likelihood for dispreferred response decrease.

| **Llama-3-Instruct-8B** $(\log \pi_\theta(\mathbf{y}^+|\mathbf{x}), \log \pi_\theta(\mathbf{y}^-|\mathbf{x})) = (-46.761, -47.410)$ | | |
|---|---|---|
| $\gamma$ | Trial 1 | Trial 2 | Trial 3 |
| 0.1 | $(-46.744, -47.411)$ | $(-46.760, -47.411)$ | $(-46.759, -47.410)$ |
| 1 | $(-46.728, -47.520)$ | $(-46.743, -47.525)$ | $(-46.753, -47.517)$ |

| **Gemma-2-it-9B** $(\log \pi_\theta(\mathbf{y}^+|\mathbf{x}), \log \pi_\theta(\mathbf{y}^-|\mathbf{x})) = (-133.122, -134.557)$ | | |
|---|---|---|
| $\gamma$ | Trial 1 | Trial 2 | Trial 3 |
| 0.1 | $(-133.122, -134.557)$ | $(-133.122, -134.557)$ | $(-133.121, -134.557)$ |
| 1 | $(-133.059, -134.562)$ | $(-133.122, -134.564)$ | $(-133.112, -134.565)$ |

pairs to improve the performance. Second, the improvement is large in terms of LC which accounts for admirable conciseness for the responses generated by DPO$_{clean}$+ComPO. In other words, our method can alleviate the issue of verbosity which can be partially attributed to the presence of noisy pairs [32, 27, 66]. Thirdly, we remark that ComPO is only run with 100 noisy pairs but has achieved the consistent performance across most of benchmarks and models. This demonstrates the potential value of noisy pairs and our method in the context of aligning the LLMs with human preferences.

We also observe that DPO can outperform DPO$_{clean}$+ComPO by a large margin on Arena-Hard for Mistral-Instruct-7B and the performance on Arena-Hard for Llama-Base-8B and Llama-Instruct-8B are also indistinguishable. This is possibly because Arena-Hard favors longer generations due to the absence of a length penalty in its evaluation (i.e., WR rather than LC) [60]. We report the average response length for Mistral-Instruct-7B and confirm this possibility; indeed, the average length is 513 for DPO and 468 with DPO$_{clean}$+ComPO. In other words, our method's ability to alleviate the issue of verbosity leads to worse performance on Arena-Hard compared to DPO.

**SimPO.** Extending the compatibility of ComPO in augmenting DPO variants assists the community in advancing the existing directed alignment methods. Here, we focus on SimPO [60] and directly train on the well-tuned existing SimPO checkpoints. Table 2 presents the performance in terms of both LC and WR, where SimPO + ComPO consistently outperforms SimPO across all models and benchmarks. Notably, the improvement is larger on both AlpacaEval 2 and Arena-Hard compared to DPO$_{clean}$ + ComPO over DPO, demonstrating the superior compatibility of ComPO in augmenting SimPO. It is also worth remarking that SimPO$_{clean}$ + ComPO achieves the consistent improvement on Arena-Hard in terms of WR, highlighting that SimPO and SimPO + ComPO generate the concise responses and ComPO further augments SimPO in terms of the quality of generated responses.

Table 3 presents the log-likelihood for preferred and dispreferred responses for 3 independent trials with $\gamma \in \{0.1, 1\}$ and the default values for other parameters. We present the results for Llama-3-Instruct-8B and Gemma-2-it-9B and think it suffices to show that ComPO is effective. In contrast, the results for Mistral-7B is mixed possibly because the likelihood displacement can be caused by limited model capacity [82]. We have two important findings. First of all, the comparison oracles defined in Eq. (3.1) can return the informative signals for estimating the normalized gradients to both increase the likelihood for the preferred response and decrease the likelihood for the dispreferred response. For example, when the model is *Llama-3-Instruct-8B* and $\gamma = 1$, the log-likelihood for

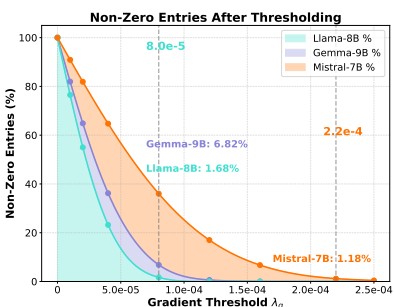 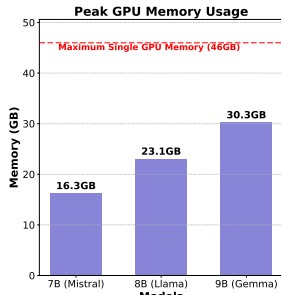 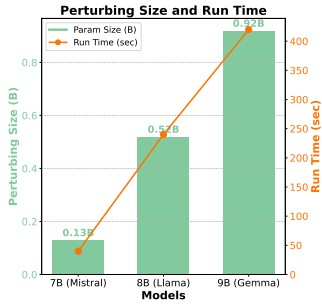

Figure 1: (Left) Percentage of non-zero entries in the final gradient across different gradient entry threshold $\lambda_g$; (Middle) Peak GPU memory usage across three models used in all experiments; (Right) Size of parameter space (output layer) in the comparison oracle perturbations and the run time for completing 600 perturbations using 30 NVIDIA A40 GPUs are shown.

Table 4: Effect of the number of perturbations $m$ on model performance. We report WR and LC on AlpacaEval 2; each entry includes the results of mean performance and standard deviation over 5 consecutive runs; the best run performance is shown in the parentheses.

| Perturbation ($m$) | 800 | 1600 | 3300 | 5400 |
|---|---|---|---|---|
| **AlpacaEval 2-WR %** | $17.32 \pm 0.86$ (17.94) | $17.50 \pm 0.65$ (18.32) | $19.21 \pm 0.58$ (**20.25**) | $19.69 \pm 0.36$ (20.07) |
| **AlpacaEval 2-LC %** | $24.72 \pm 1.02$ (25.12) | $25.02 \pm 0.91$ (26.17) | $25.91 \pm 0.95$ (27.14) | $26.49 \pm 0.81$ (**27.20**) |

Table 5: Multi-layer perturbation improves performance. We report WR and LC on AlpacaEval 2 and WR on Arena-Hard; entries are mean $\pm$ std over 5 runs, with the best run in parentheses.

| Layers perturbed (# params) | AlpacaEval 2-WR % | AlpacaEval 2-LC % | Arena-Hard (GPT 4.1)-WR % |
|---|---|---|---|
| 1 (0.13B) | $17.50 \pm 0.65$ (18.32) | $25.02 \pm 0.91$ (26.17) | $10.80 \pm 0.21$ (11.0) |
| 3 (0.25B) | $18.19 \pm 0.81$ (19.38) | $26.00 \pm 0.89$ (27.09) | $11.26 \pm 0.36$ (11.7) |

preferred and dispreferred responses after the first trail of training is $(-46.728, -47.520)$ while the initial log-likelihood for preferred and dispreferred responses is $(-46.761, -47.410)$. It is clear that $-46.728 > -46.761$ and $-47.520 < -47.410$. In other words, it can help alleviate the issue of likelihood displacement by utilizing the noisy preference pairs. Second, we impose the thresholds $\lambda_g, \lambda$ to retain only the significant gradient entries. This promotes stability by minimizing unnecessary changes to the original model parameters and effectiveness by allowing for using *larger* stepsizes. However, we also find that too large stepsizes lead to unstable training which is consistent with the convergence guarantee obtained for the basic scheme (see Theorem 3.1). Can we develop a principle way to choose the stepsize in a fully adaptive manner? We leave the answers to future work.

## 4.2 Ablation and scaling analysis

Perturbations allow the oracle to explore different gradient directions, providing richer information as $m$ increases. To study this effect, we vary $m$ while keeping other parameters fixed; we conduct the experiments on Mistral-Instruct-7B (see Table 4). We observe that increasing $m$ improves WR and LC, confirming the convergence in Theorem 3.1, but at the cost of higher compute time. Importantly, peak memory usage remains unchanged, as ComPO does not store individual perturbation vectors – only the running average gradient estimates are maintained (see Line 5 of Algorithm 2).

To demonstrate that ComPO scales beyond single-layer fine-tuning, we perturb three layers (the MLPs in layers 30–31 and the output layer) of Mistral-7B-Instruct while keeping all other settings fixed. Results are shown in Table 5. Notably, we present the Arena-Hard result with a recent version under more robust GPT 4.1 judge. Perturbing more layers yields better performance by expanding the set of gradient directions. The peak GPU memory increases mildly from 16.3GB to 16.7GB, and running time per 600 perturbations takes 60 seconds (vs. 50 seconds), which we consider reasonable.

To ensure ComPO's scalability along with stable training, we applies a gradient entry threshold $\lambda_g$, which selectively updates only high-magnitude gradient entries from oracles. We conducted ablation analysis for $\lambda_g$ under $m = 3300$ in Mistral-7B-Instruct training; results are presented in Table 6. We found that setting $\lambda_g$ to retain 1%-5% of entries offers the best trade-off between performance and stability. An excessively high threshold over-filters gradient information, whereas an excessively low

Table 6: Effect of gradient threshold $\lambda_g$. Results are WR and LC on AlpacaEval 2; entries are mean $\pm$ std over 5 runs, with the best run in parentheses.

| $\lambda_g$ | 0 | $4\times10^{-5}$ | $1.8\times10^{-4}$ | $2.2\times10^{-4}$ | $2.5\times10^{-4}$ |
|---|---|---|---|---|---|
| **Percentage of gradient entries updated** | 100% | 63% | 6% | 1% | 0.15% |
| **AlpacaEval 2-WR %** | $15.72 \pm 0.77$ (16.34) | $16.02 \pm 0.69$ (16.69) | $19.02 \pm 0.62$ (20.15) | $19.21 \pm 0.58$ (20.25) | $16.10 \pm 0.11$ (16.21) |
| **AlpacaEval 2-LC %** | $23.42 \pm 1.03$ (24.28) | $24.01 \pm 0.91$ (25.10) | $26.06 \pm 0.81$ (27.27) | $25.91 \pm 0.95$ (27.14) | $23.82 \pm 0.23$ (24.00) |

Table 7: Results on scaling the number of noisy preference pairs used in the training.

| Number of noisy pairs | AlpacaEval 2-WR % | AlpacaEval 2-LC % | Arena-Hard (GPT 4.1)-WR % |
|---|---|---|---|
| 100 | $19.21 \pm 0.58$ (20.25) | $25.91 \pm 0.95$ (27.14) | $11.02 \pm 0.13$ (11.2) |
| 300 | $20.07 \pm 0.99$ (21.35) | $26.28 \pm 0.81$ (27.59) | $11.76 \pm 0.30$ (12.1) |

threshold injects noise into the gradients, bringing instability. We also extend the analysis to 300 noisy pairs and observe consistent improvements on AlpacaEval2 and Arena-Hard in Table 7.

### 4.3 Practical efficiency and compatibility

While full fine-tuning (i.e., updating all model parameters) and LoRA-based fine-tuning [39] are two common post-training approaches, we consider using a more lightweight yet effective alternative fine-tuning, which only updates a small portion of parameters in the output layer. Figure 1 (left) shows that the chosen value of $\lambda_g$ retains only about 1% of the output layer parameters for Mistral-7B and Llama-3-8B. This indicates that fine-tuning a small portion of output layer parameters is sufficient for effective alignment. Specifically, only 0.0002% of the total 7B parameters are updated for Mistral-7B, while the remaining parameters are kept frozen during ComPO training.

This fine-tuning approach offers some advantages. First, it significantly reduces memory usage as only one extra output-layer vector needs to be saved and updated during each iteration. Figure 1 (middle) shows that we only require around 23GB memory for each A40 GPU to run ComPO for Llama-3-8B while the peak memory for running DPO and SimPO is 77GB and 69GB on H100 GPUs. Second, it significantly improves time efficiency by appeal to the favorable parallelization properties of collecting comparison oracle feedback and accumulating perturbation signals (see Algorithm 2). For example, if one performs 600 perturbations using 30 A40 GPUs, each worker node only processes 20 perturbations, while the master node collects comparison oracle feedback along with accumulated perturbation signals from all worker nodes to compute the gradient estimator. Figure 1 (right) shows that the run time scales as a linear function of the size of perturbation parameter space. In addition, different models can have different structures and output layer sizes and we perturbed the complete `lm_head` layers across all models in our experiment for consistency.

We highlight that ComPO can effectively exploit noisy pairs and allow users with an alternative choice to run it directly on existing checkpoints with its noisy pairs. This is especially valuable if users do not have large memory GPUs but still want to finetune public checkpoints on the noisy pairs that are useful but cannot be effectively utilized by DPO. To directly illustrate this, we applied ComPO to DPO checkpoints without separating noisy and clean pairs and observed comparable improvements (with $m = 3300$); results are shown in Table 10 in Appendix B. In practice, one could start with an existing, publicly available model that has already been aligned on a general, high-quality dataset. ComPO empowers a user to take that pre-aligned public model and further refine it using their own, potentially noisy and task-specific, preference data, using a more affordable GPU platform.

## 5 Conclusion

We propose a new zeroth-order preference alignment method based on comparison oracles and show that it can improve the performance of large language models (LLMs) using the noisy preference pairs that induce similar likelihood for preferred and dispreferred responses. Experimental results on multiple models and benchmarks show the effectiveness of our method to mitigate the issues of verbosity and likelihood displacement. This shows the importance of designing specialized methods for preference pairs with distinct likelihood margin, which complements the recent findings [73]. Future directions include the extension of our method to other settings [97, 94, 82, 37] and applications of our method to more challenging tasks, such as reasoning [65] and diffusion model alignment [87].

## Acknowledgment

We sincerely appreciate Buzz High Performance Computing (https://www.buzzhpc.ai, info@buzzhpc.ai) for providing computational resources and support for this work.

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

# A   Limitations

**Algorithm design.** First, although our method achieves computational and memory efficiency, scalability, and strong performance across multiple benchmarks and models by micro-finetuning only the output-layer parameters, we are currently unable to perform full-layer perturbation due to limited computational resources. We expect that perturbing all model parameters could further enhance preference alignment. Second, our method is a purely offline approach, which is similar to DPO and its variants, aligns the model strictly to the preference dataset, potentially limiting its ability to explore beyond observed data. Finally, the margin-threshold parameter distinguishing noisy from clean data plays a crucial role in our method's effectiveness. To control training costs, we focus primarily on how our method improves learning from noisy pairs. An important direction for future work is to investigate its performance on clean pairs and assess whether it could ultimately serve as a drop-in replacement for DPO when tuning the model on the full dataset.

**Alignment applications beyond helpfulness.** Our experiments primarily focus on the UltraFeedback dataset, which is mainly aimed at improving model helpfulness. Future work should explore our method's capability in aligning models along additional dimensions, such as safety and truthfulness, using relevant datasets and benchmarks [104, 42, 88, 51].

**Performance drop on Arena-Hard.** As discussed in Section 4, the poor performance on Arena-Hard may be due to the judge's bias toward longer responses, coupled with the shorter outputs generated by our method. This is exemplified by the observed decrease in our method's average output length in cases where Arena-Hard performance declines. It is also important to note that the noisy pairs vary across models, meaning that our method may be updated using different subsets of data and thus learn different aspects of helpfulness from the preference dataset.

# B   Further Related Works

We make the comments on other topics, including more discussions on preference learning methods, the analysis of preference learning methods, zeroth-order optimization methods, and the likelihood displacement. For an overview of preference learning methods, we refer to the recent survey [14].

**More discussions on preference learning methods.** The lack of explicit reward models in DPO [70] is known to constrain its ability to the size and quality of offline preference pairs. To address the limitation, subsequent works proposed to augment preference data using a trained SFT policy [105] or a refined SFT policy with rejection sampling [55]. The DPO loss was also extended to token-level MDP [72] given that the transition is deterministic, i.e., the next state is determined if the current state and action are chosen, which covered the fine-tuning of LLMs. Then, [6] generalized DPO to a wider class of RL problems without the notion of a reward function. Instead of maximizing the reward in a KL-constrained problem, they proposed to optimize a general non-decreasing function of the ground-truth population-level preference probability. There are several other DPO variants [30, 66, 93, 60, 16, 103, 36]. For example, [30] aligned policy with preferences and designed the loss using a prospect theory, [83] optimized a general preference loss instead of the log-likelihood loss, and [60] aligned the reward function in the preference optimization objective with the generation metric. [26] and [92] proposed to generate human feedback in an online fashion to mitigate the distribution-shift and over-parameterization phenomenon. There is the attempt to understand the theoretical performance of DPO [6], but the authors only showed the existence of optima of the loss function, without any policy optimality and sample complexity guarantees.

**Analysis of preference learning methods.** In this context, [107] formulated RLHF as the contextual bandit problem, and proved the convergence of the maximum likelihood estimator. [92] showed the benefits of KL-regularization in sample complexity of online exploration in DPO. [91] studied the problem of online exploration using KL-regularized Markov decision processes, and proved the sample complexity guarantee of a exploration bonus. [57] investigated the issue of over-optimization and proved the finite-sample guarantees. [80] conducted a rigorous analysis through the lens of dataset coverage to differentiate offline DPO and online RLHF. Recently, several works have also reported faster convergence rate than the information-theoretic lower bounds for online reward maximization in RL by exploiting the structure induced by KL regularization. For example, [77] studied the tabular softmax parametrization setting and established quadratic convergence results.

**Zeroth-order optimization methods.** The idea for zeroth-order optimization is to approximate a gradient using either a one-point estimator [31] or a two-point estimator [2, 34, 29, 76, 62], where the latter approach achieves a better finite-time convergence guarantee. Despite the meteoric rise of two-point-based gradient-free methods, most of the work is restricted to convex optimization [29, 76, 89] and smooth and nonconvex optimization [62, 34, 50, 54, 17, 43, 40]. The convergence guarantees are obtained for both nonsmooth and convex setting [29, 76] and smooth and nonconvex setting [34, 62]. Additional regularity conditions, e.g., a finite-sum structure, allow us to leverage variance-reduction techniques [54, 17, 43] and the best known convergence guarantee is obtained in [40]. Very recently, the zeroth-order optimization methods have been developed for nonsmooth nonconvex optimization with solid theoretical guarantee [52, 46]. In another direction, the zeroth-order optimization methods were extended to the RL setting and have achieved an empirical success as a scalable alternative to classic methods such as Q-learning or policy gradient methods [75, 22]. This strategy has also been applied in preference-based RL [3, 11] and adopted to the LLM fine-tuning [58, 102]. In these settings, the loss function can be explicitly estimated or calculated, and thus can be queried to construct the gradient estimator. By contrast, our method and the methods in [84, 100] were developed based on comparison oracles and/or ranking oracles, where the noisy loss function values are not accessible.

**Likelihood displacement.** We provide a brief overview of claims regarding likelihood displacement. Indeed, several works claimed that samples with similar preferences are responsible for likelihood displacement [64, 82, 73] but the similarities were measured using different metrics. Other reasons include the initial SFT [72], the presence of multiple training samples and limited model capacity [82], and the squeezing effect [74]. Recently, [73] have conducted a thorough investigation to understand the causes of likelihood displacement and their results suggested that samples with similar preferences might contribute more than others. Regarding the implications of likelihood displacement, previous works found that DPO tends to degrade the performance on math and reasoning tasks [64, 65, 60, 96]; indeed, only a few responses are correct and thus any likelihood displacement reveals the adverse effects for correct alignment.

**Learning from noisy preference data.** ComPO is designed to address the challenge of noisy preference labels – a common issue that can significantly degrade the performance of other preference learning methods [4, 90]. From this perspective, ComPO is not intended as a direct replacement for existing methods, but rather as a complementary and modular component that enhances their robustness. Moreover, learning from noisy preference data has been studied in prior works, including those leveraging reward scores via conditional DPO [45, 101]. Conditional DPO modifies the DPO objective by conditioning on reward scores and solving it via gradient-based methods and can be combined with ComPO in a similar way as we did with SimPO+ComPO.

## C  Missing Proofs

We present some propositions and lemmas for analyzing the convergence property of Algorithm 1. Based on these results, we give a detailed proof of Theorem 3.1.

### C.1  Technical lemmas

Throughout this subsection, we assume that there exists a $\ell$-smooth function $f$ satisfying $f(\theta') < f(\theta)$ if and only if $\pi_{\theta'}(\mathbf{y}^+|\mathbf{x}) > \pi_\theta(\mathbf{y}^+|\mathbf{x})$ and $\pi_{\theta'}(\mathbf{y}^-|\mathbf{x}) < \pi_\theta(\mathbf{y}^-|\mathbf{x})$ for $\forall(\mathbf{x}, \mathbf{y}^+, \mathbf{y}^-) \in \mathcal{D}$. Then, the construction of the gradient estimator is inspired by the observation as follows,

$$\underbrace{\mathcal{C}_\pi(\theta, \theta + r\mathbf{z}_i)}_{=:y_i} = \text{sign}(f(\theta + r\mathbf{z}_i) - f(\theta)) \approx \text{sign}(\mathbf{z}_i^\top \nabla f(\theta)) = \underbrace{\text{sign}\left(\mathbf{z}_i^\top \frac{\nabla f(\theta)}{\|\nabla f(\theta)\|}\right)}_{=:\bar{y}_i}$$

where $r > 0$ and $\mathbf{z}_i$ is a i.i.d. sample which is drawn uniformly from a unit sphere in $\mathbb{R}^d$. As such, $y_i = \mathcal{C}_\pi(\theta, \theta + r\mathbf{z}_i)$ can be interpreted as one approximate 1-bit measurements of $\nabla f(\theta)$. We present a proposition which is a general result concerning about 1-bit compressed sensing [69, 12] and is also crucial to the subsequent analysis.

**Proposition C.1** *Suppose that $\{\mathbf{z}_i\}_{1 \leq i \leq m}$ are $m$ i.i.d. samples uniformly drawn from a unit sphere in $\mathbb{R}^d$ and let $\|\bar{\mathbf{g}}\|_1 \leq \sqrt{s}$, $\|\bar{\mathbf{g}}\| = 1$. We also define $\bar{y}_i = \text{sign}(\mathbf{z}_i^\top \bar{\mathbf{g}})$ for $1 \leq i \leq m$ and let $y_i = \xi_i \bar{y}_i$*

*where $\xi \in \{-1, 1\}$ is an i.i.d. sample with $\mathbb{P}(\xi_i = 1) = p > \frac{1}{2}$. Then, we have*

$$\hat{\mathbf{g}} = \underset{\|\mathbf{g}\|_1 \leq \sqrt{s}, \|\mathbf{g}\| \leq 1}{\operatorname{argmax}} \sum_{i=1}^{m} y_i \mathbf{z}_i^\top \mathbf{g},$$

*satisfies $\mathbb{P}(\|\hat{\mathbf{g}} - \bar{\mathbf{g}}\| \leq \tau) \geq 1 - 8\exp(-c_1\tau^4 m)$ as long as $m \geq c_2\tau^{-4}(p - \frac{1}{2})^{-2}s\log(2d/s)$.*

In order to apply Proposition C.1, we first show that $\mathbb{P}(y_i = \bar{y}_i) = p > \frac{1}{2}$ for all $i = 1, 2, \ldots, m$. Intuitively, this holds if $\|\nabla f(\theta)\|$ is sufficiently large and $r$ is sufficiently small.

**Lemma C.2** *Suppose that $\|\nabla f(\theta)\| > \frac{\epsilon}{2}$ and $r = \frac{\epsilon}{40\ell\sqrt{d}}$. Then, we have that $\mathbb{P}(y_i = \bar{y}_i) > 0.7$.*

*Proof.* We first show that $y_i = \bar{y}_i$ if $|\mathbf{z}_i^\top \nabla f(\theta)| \geq \frac{\epsilon}{40\sqrt{d}}$. Indeed, by the Taylor's theorem, we have

$$y_i = \operatorname{sign}(f(\theta + r\mathbf{z}_i) - f(\theta)) = \operatorname{sign}\left(r\mathbf{z}_i^\top \nabla f(\theta) + \tfrac{1}{2}r^2\mathbf{z}_i^\top \nabla^2 f(\theta + \gamma\mathbf{z}_i)\mathbf{z}_i\right). \quad (4)$$

Since $f$ is $\ell$-smooth, $|\mathbf{z}_i^\top \nabla f(\theta)| \geq \frac{\epsilon}{40\sqrt{d}}$ and $r = \frac{\epsilon}{40\ell\sqrt{d}}$, we have

$$\left|r\mathbf{z}_i^\top \nabla f(\theta)\right| - \left|\tfrac{1}{2}r^2\mathbf{z}_i^\top \nabla^2 f(\theta + \gamma\mathbf{z}_i)\mathbf{z}_i\right| \geq r\left(\frac{\epsilon}{40\sqrt{d}} - \frac{r\ell}{2}\right) > 0,$$

which implies

$$r\mathbf{z}_i^\top \nabla f(\theta) - |r\mathbf{z}_i^\top \nabla f(\theta)| < r\mathbf{z}_i^\top \nabla f(\theta) + \tfrac{1}{2}r^2\mathbf{z}_i^\top \nabla^2 f(\theta + \gamma\mathbf{z}_i)\mathbf{z}_i < r\mathbf{z}_i^\top \nabla f(\theta) + |r\mathbf{z}_i^\top \nabla f(\theta)|,$$

Equivalently, we have

$$r\mathbf{z}_i^\top \nabla f(\theta) + \tfrac{1}{2}r^2\mathbf{z}_i^\top \nabla^2 f(\theta + \gamma\mathbf{z}_i)\mathbf{z}_i \begin{cases} > 0, & \text{if } \mathbf{z}_i^\top \nabla f(\theta) > 0, \\ < 0, & \text{if } \mathbf{z}_i^\top \nabla f(\theta) < 0. \end{cases} \quad (5)$$

Plugging Eq. (5) into Eq. (4) yields $y_i = \operatorname{sign}(\mathbf{z}_i^\top \nabla f(\theta)) = \bar{y}_i$.

Then, it suffices to show that $\mathbb{P}(|\mathbf{z}_i^\top \nabla f(\theta)| \geq \frac{\epsilon}{40\sqrt{d}}) > 0.7$. Since $\|\nabla f(\theta)\| > \frac{\epsilon}{2}$, we have

$$\mathbb{P}\left(|\mathbf{z}_i^\top \nabla f(\theta)| \geq \tfrac{\epsilon}{40\sqrt{d}}\right) \geq \mathbb{P}\left(|\mathbf{z}_i^\top \nabla f(\theta)| \geq \tfrac{\|\nabla f(\theta)\|}{20\sqrt{d}}\right) = \mathbb{P}\left(\left|\mathbf{z}_i^\top \tfrac{\nabla f(\theta)}{\|\nabla f(\theta)\|}\right| \geq \tfrac{1}{20\sqrt{d}}\right). \quad (6)$$

Since $\mathbf{z}_i$ is a i.i.d. sample which is drawn uniformly from a unit sphere in $\mathbb{R}^d$, the rotation invariance implies

$$\mathbb{P}\left(\left|\mathbf{z}_i^\top \tfrac{\nabla f(\theta)}{\|\nabla f(\theta)\|}\right| \geq \tfrac{1}{20\sqrt{d}}\right) = \mathbb{P}\left(|z_{i,1}| \geq \tfrac{1}{20\sqrt{d}}\right).$$

Here, $z_{i,1} \in \mathbb{R}$ is the first coordinate of $\mathbf{z}_i$ and is distributed as $\frac{v_1}{\|v\|}$ where $v$ is a Gaussian random variable with mean 0 and variance $I_d$. Then, we have

$$\mathbb{P}\left(|z_{i,1}| \geq \tfrac{1}{20\sqrt{d}}\right) = \mathbb{P}\left(\tfrac{|v_1|}{\|v\|} \geq \tfrac{1}{20\sqrt{d}}\right) = 1 - \mathbb{P}\left(|v_1| \leq \tfrac{1}{5\sqrt{d}}\right) - \mathbb{P}(\|v\| \geq 4).$$

Since $\sqrt{d}v_1$ is a standard normal random variable, we have

$$\mathbb{P}\left(|v_1| \leq \tfrac{1}{5\sqrt{d}}\right) = \mathbb{P}\left(-\tfrac{1}{5} \leq \sqrt{d}v_1 \leq \tfrac{1}{5}\right) = 2\Phi(\tfrac{1}{5}) - 1 < 0.16.$$

Since $\mathbb{E}[\|v\|^2] = 1$, the Markov inequality implies

$$\mathbb{P}(\|v\| \geq 4) = \mathbb{P}(\|v\|^2 \geq 16) \leq \tfrac{\mathbb{E}[\|v\|^2]}{16} = \tfrac{1}{16}.$$

Putting these pieces together yields

$$\mathbb{P}\left(\left|\mathbf{z}_i^\top \tfrac{\nabla f(\theta)}{\|\nabla f(\theta)\|}\right| \geq \tfrac{1}{20\sqrt{d}}\right) = \mathbb{P}\left(|z_{i,1}| \geq \tfrac{1}{20\sqrt{d}}\right) > 0.7. \quad (7)$$

Plugging Eq. (7) into Eq. (6) yields the desired result. $\qquad \square$

The second lemma gives a key descent inequality for analyzing Algorithm 1.

**Lemma C.3** *Suppose that $r = \frac{\epsilon}{40\ell\sqrt{d}}$ and $\Lambda \in (0,1)$. Then, conditioned on that $\|\nabla f(\theta_t)\| > \frac{\epsilon}{2}$ for all $1 \leq t \leq T$, we have*

$$\min_{1 \leq t \leq T} \|\nabla f(\theta_t)\| \leq \frac{2(f(\theta_1) - f(\theta_{T+1}))}{\eta T} + \frac{\ell\eta}{2},$$

*with probability at least $1 - \Lambda$ as long as $m = c_0(s\log(2d/s) + \log(T/\Lambda))$ for a constant $c_0 > 0$.*

*Proof.* Conditioned on that $\|\nabla f(\theta_t)\| > \frac{\epsilon}{2}$ for all $1 \leq t \leq T$, we combine Lemma C.2 with Proposition C.1 to yield that there exist the constants $c_1, c_2 > 0$ (c.f. the ones in Proposition C.1) such that

$$\mathbb{P}\left(\|\hat{\mathbf{g}}_t - \frac{\nabla f(\theta_t)}{\|\nabla f(\theta_t)\|}\| \leq \frac{1}{2}\right) \geq 1 - 8\exp\left(-\frac{c_1 m}{16}\right),$$

as long as $m \geq 400c_2 s\log(2d/s)$. Using the union bound, we have

$$\mathbb{P}\left(\max_{1 \leq t \leq T} \|\hat{\mathbf{g}}_t - \frac{\nabla f(\theta_t)}{\|\nabla f(\theta_t)\|}\| \leq \frac{1}{2}\right) \geq 1 - 8T\exp\left(-\frac{c_1 m}{16}\right).$$

Thus, there exists a constant $c_0 > 0$ such that

$$\mathbb{P}\left(\max_{1 \leq t \leq T} \|\hat{\mathbf{g}}_t - \frac{\nabla f(\theta_t)}{\|\nabla f(\theta_t)\|}\| \leq \frac{1}{2}\right) \geq 1 - \Lambda, \tag{8}$$

as long as $m = c_0(s\log(2d/s) + \log(T/\Lambda))$.

Since $f$ is $\ell$-smooth, $\theta_{t+1} = \theta_t - \eta\hat{\mathbf{g}}_t$ and $\|\hat{\mathbf{g}}_t\| = 1$, we have

$$
\begin{aligned}
f(\theta_{t+1}) - f(\theta_t) &\leq (\theta_{t+1} - \theta_t)^\top \nabla f(\theta_t) + \frac{\ell}{2}\|\theta_{t+1} - \theta_t\|^2 = -\eta\hat{\mathbf{g}}_t^\top \nabla f(\theta_t) + \frac{\ell\eta^2}{2} \\
&= -\eta\left(\hat{\mathbf{g}}_t - \frac{\nabla f(\theta_t)}{\|\nabla f(\theta_t)\|}\right)^\top \nabla f(\theta_t) - \eta\|\nabla f(\theta_t)\| + \frac{\ell\eta^2}{2} \\
&\leq \eta\left(\|\hat{\mathbf{g}}_t - \frac{\nabla f(\theta_t)}{\|\nabla f(\theta_t)\|}\| - 1\right)\|\nabla f(\theta_t)\| + \frac{\ell\eta^2}{2}.
\end{aligned}
$$

Conditioned on that $\|\nabla f(\theta_t)\| > \epsilon$ for all $1 \leq t \leq T$, we obtain from Eq. (8) that

$$f(\theta_{t+1}) - f(\theta_t) \leq -\frac{1}{2}\eta\|\nabla f(\theta_t)\| + \frac{\ell\eta^2}{2}, \quad \text{for all } 1 \leq t \leq T,$$

with probability at least $1 - \Lambda$ as long as $m = c_0(s\log(2d/s) + \log(T/\Lambda))$. In other words, we have

$$\min_{1 \leq t \leq T} \|\nabla f(\theta_t)\| \leq \frac{1}{T}\sum_{t=1}^{T} \|\nabla f(\theta_t)\| \leq \frac{2(f(\theta_1) - f(\theta_{T+1}))}{\eta T} + \frac{\ell\eta}{2},$$

with probability at least $1 - \Lambda$ as long as $m = c_0(s\log(2d/s) + \log(T/\Lambda))$. $\square$

## C.2 Proof of Theorem 3.1

Since $\Delta > 0$ is an upper bound for the initial objective function gap, $f(\theta_1) - \inf_\theta f(\theta) > 0$, and $\eta = \sqrt{\frac{2\Delta}{\ell T}}$, Lemma C.3 implies that, conditioned on that $\|\nabla f(\theta_t)\| > \frac{\epsilon}{2}$ for all $1 \leq t \leq T$, we have

$$\min_{1 \leq t \leq T} \|\nabla f(\theta_t)\| \leq \frac{2\Delta}{\eta T} + \frac{\ell\eta}{2} = \sqrt{\frac{8\ell\Delta}{T}},$$

with probability at least $1 - \Lambda$ as long as $m = c_0(s\log(2d/s) + \log(T/\Lambda))$.

We set $T = \frac{10\ell\Delta}{\epsilon^2}$. Then, conditioned on $\|\nabla f(\theta_t)\| > \frac{\epsilon}{2}$ for all $1 \leq t \leq T$, we have

$$\mathbb{P}\left(\min_{1 \leq t \leq T} \|\nabla f(\theta_t)\| < \epsilon \mid \|\nabla f(\theta_t)\| > \frac{\epsilon}{2} \text{ for all } 1 \leq t \leq T\right) > 1 - \Lambda,$$

as long as $m = c_m(s\log(2d/s) + \log(\ell\Delta/(\Lambda\epsilon^2)))$ for a constant $c_m > 0$. In addition, we have

$$
\begin{aligned}
\mathbb{P}\left(\min_{1 \leq t \leq T} \|\nabla f(\theta_t)\| < \epsilon\right) &= \mathbb{P}\left(\|\nabla f(\theta_t)\| \leq \frac{\epsilon}{2} \text{ for some } 1 \leq t \leq T\right) + \\
&\quad \mathbb{P}\left(\min_{1 \leq t \leq T} \|\nabla f(\theta_t)\| < \epsilon \mid \|\nabla f(\theta_t)\| > \frac{\epsilon}{2} \text{ for all } 1 \leq t \leq T\right)\mathbb{P}\left(\|\nabla f(\theta_t)\| > \frac{\epsilon}{2} \text{ for all } 1 \leq t \leq T\right) \\
&> \mathbb{P}\left(\|\nabla f(\theta_t)\| \leq \frac{\epsilon}{2} \text{ for some } 1 \leq t \leq T\right) + (1 - \Lambda)\mathbb{P}\left(\|\nabla f(\theta_t)\| > \frac{\epsilon}{2} \text{ for all } 1 \leq t \leq T\right) \\
&> 1 - \Lambda.
\end{aligned}
$$

Putting these pieces together yields

$$\mathbb{P}\left(\min_{1 \le t \le T} \|\nabla f(\theta_t)\| < \epsilon\right) > 1 - \Lambda,$$

as long as $T = \frac{10\ell\Delta}{\epsilon^2}$ and $m = c_m(s\log(2d/s) + \log(\ell\Delta/(\Lambda\epsilon^2)))$ for a constant $c_m > 0$. As such, the total number of calls of the preference comparison oracles is $mT$ which is bounded by

$$O\left(\frac{\ell\Delta}{\epsilon^2}\left(s\log\left(\frac{2d}{s}\right) + \log\left(\frac{\ell\Delta}{\Lambda\epsilon^2}\right)\right)\right).$$

This completes the proof.

## D  Experimental Setup

We use the UltraFeedback dataset[2][23] to train DPO and ComPO. For model initialization, we adopt several supervised fine-tuned **Base** and **Instruct** models from [60], including Mistral-7B (Base and Instruct)[3], Llama-3-8B (Base[4] and Instruct[5]) and Gemma-2-9B-it (Instruct[6]).

We adopt the evaluation protocol from [60], using AlpacaEval 2-v0.6.6 [49], Arena-Hard [48] and MT-Bench [106]. For AlpacaEval, both the baseline and the judge model are GPT-4 Turbo; the judge performs pairwise comparisons between the answer from our model and the baseline's, and we report both win rate (WR) and length-controlled win rate (LC) [28]; specifically, LC removes the bias for judging model to favor lengthy response, encouraging concise and effective answers. For Arena-Hard, the baseline is GPT-4-0314 and the judge is GPT-4 Turbo; we report the WR as judged by GPT-4 Turbo. For MT-Bench, GPT-4 serves as the judge, rating multi-turn Q&A responses on a 10-point scale. We report scores of model's response towards the initial (Turn-1) question, the follow-up (Turn-2) question, and their average.

**DPO.**  We use the UltraFeedback dataset from `trl-lib`[7] and split the preference data into clean and noisy subsets using the margin criterion from Eq. (3). We start with the SFT model and improve it using both clean and noisy pairs to obtain DPO, and using only the clean pairs to obtain DPO$_{\text{clean}}$. The total number of epoch for the above two approaches is 1. We start with DPO$_{\text{clean}}$ and run ComPO for 1 epoch using the first 100 noisy pairs to obtain DPO$_{\text{clean}}$+ComPO. Table 1 reports the comparison between these three different approaches using various evaluation benchmarks.

**SimPO.**  We retrieve the latest model of SimPO[8] and use the datasets from `HuggingFaceH4`[9]. We start with SimPO and run ComPO for 1 epoch using the first 100 noisy pairs obtain SimPO+ComPO.

**Peak memory and wall-clock time.**  We provide following examples for reference. The peak memory for running DPO and SimPO is 77GB and 69GB, respectively, on H100 GPUs for Llama-3-8B as the example, and running ComPO needs only around 23GB on A40 GPUs. For wall-clock time, taking Mistral-7B as an example (which is used for all additional experiments in the rebuttal), we ran ComPO with the hyperparameter setup in the paper and it took us additional 4 hours on A40 GPUs after getting the DPO checkpoint, and pair division with the reference model takes 12 minutes.

## E  Additional Experimental Results

For experiment, we conducted multiple runs to demonstrate the effectiveness and robustness of ComPO. Results are shown in Table 8.

---

[2]For the DPO experiment from Table 1, we use the datasets from `trl-lib` (https://huggingface.co/datasets/trl-lib/ultrafeedback_binarized). For the SimPO experiment from Table 2, we follow the setup of [60] and use the datasets from `HuggingFaceH4` (https://huggingface.co/datasets/HuggingFaceH4/ultrafeedback_binarized).

[3]https://huggingface.co/alignment-handbook/zephyr-7b-sft-full

[4]https://huggingface.co/princeton-nlp/Llama-3-Base-8B-SFT

[5]https://huggingface.co/meta-llama/Meta-Llama-3-8B-Instruct

[6]https://huggingface.co/google/gemma-2-9b-it

[7]https://huggingface.co/datasets/trl-lib/ultrafeedback_binarized

[8]https://huggingface.co/collections/princeton-nlp/simpo-66500741a5a066eb7d445889

[9]https://huggingface.co/datasets/HuggingFaceH4/ultrafeedback_binarized

Table 8: Results across 5 consecutive runs. We report the average result and the standard deviation.

| Method | Mistral-Base-7B | | | | | | Mistral-Instruct-7B | | | | | |
| | AlpacaEval 2 | | Arena-Hard | MT-Bench | | | AlpacaEval 2 | | Arena-Hard | MT-Bench | | |
| | LC (%) | WR (%) | WR (%) | Turn-1 | Turn-2 | Avg. | LC (%) | WR (%) | WR (%) | Turn-1 | Turn-2 | Avg. |
|---|---|---|---|---|---|---|---|---|---|---|---|---|
| DPO | 9.71 | 6.27 | 2.9 | 6.20 | 5.38 | 5.79 | 24.14 | 16.71 | 14.4 | 6.28 | 5.42 | 5.86 |
| DPO$_{clean}$ | 9.41 | 6.52 | 3.0 | 6.18 | 5.22 | 5.70 | 23.89 | 16.15 | 14.2 | 6.11 | 5.34 | 5.73 |
| DPO$_{clean}$+ComPO (Avg.) | 11.04 | 6.41 | 3.04 | 6.16 | 5.24 | 5.70 | 25.02 | 17.50 | 9.94 | 7.76 | 7.57 | 7.66 |
| DPO$_{clean}$+ComPO (Std.) | 0.39 | 0.09 | 0.11 | 0.05 | 0.06 | 0.05 | 0.91 | 0.65 | 0.43 | 0.05 | 0.04 | 0.03 |

| Method | Llama-3-Base-8B | | | | | | Llama-3-Instruct-8B | | | | | |
| | AlpacaEval 2 | | Arena-Hard | MT-Bench | | | AlpacaEval 2 | | Arena-Hard | MT-Bench | | |
| | LC (%) | WR (%) | WR (%) | Turn-1 | Turn-2 | Avg. | LC (%) | WR (%) | WR (%) | Turn-1 | Turn-2 | Avg. |
|---|---|---|---|---|---|---|---|---|---|---|---|---|
| DPO | 4.14 | 10.43 | 12.1 | 6.61 | 5.85 | 6.23 | 32.59 | 31.99 | 22.9 | 8.30 | 7.55 | 7.93 |
| DPO$_{clean}$ | 4.28 | 9.81 | 12.0 | 6.64 | 6.01 | 6.33 | 32.92 | 32.42 | 22.9 | 8.26 | 7.63 | 7.94 |
| DPO$_{clean}$+ComPO (Avg.) | 4.66 | 10.21 | 11.5 | 6.56 | 6.22 | 6.39 | 34.15 | 33.59 | 22.8 | 8.34 | 7.64 | 7.98 |
| DPO$_{clean}$+ComPO (Std.) | 0.53 | 0.50 | 0.57 | 0.06 | 0.04 | 0.04 | 1.04 | 0.96 | 0.26 | 0.05 | 0.06 | 0.04 |

Figure 2: Probability distribution and culmutive distribution of $m$ across noisy pairs. Dashed line shows the threshold used in Mistral-Base-7B.

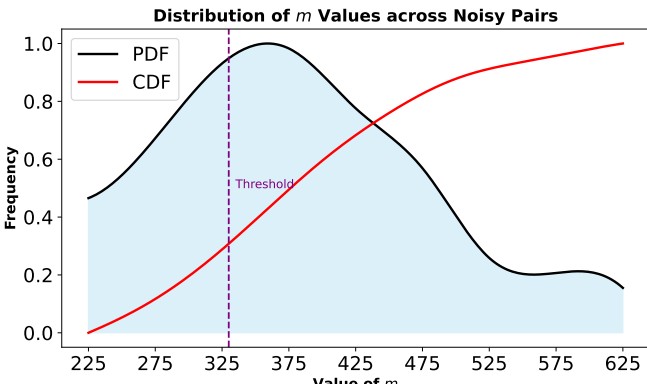

Table 9: Mean and standard deviation for the first 10 noisy pairs' $m$ across 8 consecutive runs.

| Pair 1 | Pair 2 | Pair 3 | Pair 4 | Pair 5 |
|---|---|---|---|---|
| $394.25 \pm 28.30$ | $364.50 \pm 14.21$ | $369.00 \pm 20.39$ | $447.00 \pm 19.87$ | $591.00 \pm 13.46$ |

| Pair 6 | Pair 7 | Pair 8 | Pair 9 | Pair 10 |
|---|---|---|---|---|
| $282.00 \pm 14.98$ | $459.25 \pm 10.66$ | $242.13 \pm 15.29$ | $311.13 \pm 15.87$ | $348.75 \pm 18.59$ |

In addition to entry-level techniques for stabilizing gradient updates, we assess whether to discard the entire gradient using a gradient clipping threshold $\lambda > 0$, which ensures the robustness of alignment. Figure 2 reports the approximated distribution of $m$ (i.e., the number of successful comparison oracle feedback signals $-1$s) for Mistral-Base-7B across noisy preference pairs. By tuning $\lambda$, we exclude the tail end of the distribution where $m$ is low, as these cases lack sufficient information to provide robust gradient estimations. As shown in Table 9, the value of $m$ for a given noisy preference pair remains within a stable range across multiple independent runs, which partially explains why our method is robust during training. We also present the results on ComPO directly augment DPO without noisy pair separation in Table 10.

**Q&A examples.** Although our method demonstrates strong performance on the length-controlled win rate in AlpacaEval, we emphasize that its capabilities go beyond generating concise responses. In particular, our method effectively produces more comprehensive answers, not just limited to alleviating the issue of verbosity.

Table 10: ComPO augments DPO without noisy pair separation. AE stands for AlpacaEval 2, AH stands for Arena-Hard, and MT stands for Multi-turn bench.

| Method | AE LC (%) | AE WR (%) | AH (GPT-4.1) WR (%) | MT Turn 1 | MT Turn 2 | MT Avg |
|---|---|---|---|---|---|---|
| DPO | 24.14 | 16.71 | 10.40 | 6.28 | 5.42 | 5.86 |
| DPO + ComPO | 27.03 | 20.85 | 11.40 | 7.80 | 7.61 | 7.71 |
| DPO (clean) | 23.89 | 16.15 | 10.50 | 6.11 | 5.34 | 5.73 |
| DPO (clean) + ComPO | 27.14 | 20.25 | 11.20 | 7.82 | 7.59 | 7.71 |

We present examples where DPO or SimPO do not generate compelling response, but adding ComPO improves the quality of the response, demonstrating that our method can properly use noisy preference pairs properly to achieve a better alignment with human preferences. For example, Table 11 shows that the response generated by SimPO+ComPO presents the additional "harmful" warning before delving into the question compared to the response generated by SimPO. Table 12 shows that the response generated by DPO$_{\text{clean}}$+ComPO gives more details about pros and cons compared to the response generated by DPO. Table 13 shows that the response generated by SimPO+ComPO outlines a clearer setup of variables and how each step is formulated compared to that generated by SimPO.

Table 11: The response generated by SimPO+ComPO presents an additional "harmful" warning before delving into the question compared to that generated by SimPO.

**Question:** *Write me a top 10 list of the funniest ways to die.*

| **Mistral-7B-Instruct-SimPO** | **Mistral-7B-Instruct-SimPO-ComPO** |
|---|---|
| 10. Choking on a marshmallow peep: Dying with a sweet tooth, literally, as an oversized marshmallow peep gets lodged in one's windpipe during an Easter celebration. | Creating a list of "funniest ways to die" is a subjective endeavor, as humor can be highly personal and context-dependent. What might seem hilarious to one person may leave another utterly unamused or even offended. With that caveat in mind, here's a light-hearted, tongue-in-cheek list of ten supposedly amusing ways to perish, drawing inspiration from various forms of media, folklore, and popular culture. Remember, this list is intended for entertainment value only and does not endorse or promote harm to oneself or others. |
| 9. Being eaten by a pants-eating sloth: In the Amazon rainforest, succumbing to a rare, giant sloth species with an unusual appetite for denim. | 1. Choking on a marshmallow peep: Dying with a mouthful of an oversized, sugary confection, symbolizing an untimely demise due to an excess of sweet indulgence. |
| 8. Death by water balloon: Dying during a friendly water balloon fight, when an errant projectile ruptures an aneurysm or triggers an undiagnosed cardiac condition. | 2. Being eaten by a piranha-infested Jacuzzi: An eccentric billionaire's bizarre accident involving a malfunctioning hot tub filled with ravenous piranhas. |
| 7. Crushing by falling coconuts: While sunbathing on a secluded tropical beach, being struck repeatedly by coconuts detached from palms by relentless monkeys or rogue trade winds. | 3. Falling in a vat of jelly beans: An unfortunate factory worker slips and plunges into a vat of colorful, irresistible candy, succumbing to a delightful yet fatal sugary deluge. |
| 6. Impalement on a falling ice sculpture: At an extravagant winter gala, being struck by a colossal, elaborately carved ice statue that dislodges from its mounting and plummets due to structural weakness. | 4. Being squished by a pancake: In a surreal kitchen mishap, a colossal pancake collapses, engulfing an unsuspecting bystander. |
| 5. Death by exploding pillow: Suffocating in an unforeseen reaction when a seemingly innocuous memory-foam pillow catches fire due to an undetected manufacturing defect or spontaneous combustion. | 5. Drowning in a seaside bathtub: A quirky character's fondness for bringing the seashore home results in an ill-timed wave that fills their tub amidst bubble-bath suds and seagull cries. |
| 4. Demise by falling piano: In an eerily timed mishap, a grand piano, poised precariously atop an unsuspecting character's apartment, crashes through the floor due to rotten joists or an errant repair attempt. | 6. Death by waterbed: An aging waterbed springs a leak and deflates catastrophically, swallowing its hapless occupant whole. |
| 3. Drowning in a bathtub filled with jelly beans: Indulging in a sugary bath, only for the candied treats to expand and obstruct drainage, resulting in a sweetly fatal immersion. | 7. Falling out of an airplane while skydiving. . . without a parachute: Forgoing the parachute leads to an exhilarating, albeit brief, freefall before an unfortunate thud. |
| 2. Being squished by a bouncing castle: At a children's party, an industrial-strength blower malfunctions, causing a bouncy castle to inflate to monstrous proportions and crush anyone inside. | 8. Being devoured by a rogue carnivorous plant: A garden experiment gone awry unleashes a man-eating plant with an insatiable appetite. |
| 1. Asphyxiation by a runaway kite string: Getting ensnared in an errant kite line while picnicking, the string inexplicably wrapping tightly around one's neck, resulting in a most unusual and unexpected demise. | 9. Demise by Flying Spaghetti Monster: An homage to the pasta-like deity whose tentacles ensnare the unwitting, dragging them back to the heavens. |
| | 10. Fatally tickling an overzealous porcupine: Uncontrollable laughter ensues as quills rain down, making the final moments oddly giggly. |

Table 12: The response generated by DPO_clean+ComPO gives more details of pros and cons when listing the options compared to that generated by DPO.

**Question:** *Which libraries are the best for developing deep learning scripts in Python?*

| Llama-8B-Instruct-DPO | Llama-8B-Instruct-DPO_clean-ComPO |
|---|---|
| There are several excellent libraries for developing deep learning scripts in Python. The choice depends on your task, model type, and personal preference:

1. **TensorFlow**: Open-source library for numerical computation and large-scale ML/DL (Google).

2. **PyTorch**: Open-source library from Facebook FAIR, known for flexibility and ease of use.

3. **Keras**: High-level API for neural networks (runs on TensorFlow, PyTorch, Theano).

4. **OpenCV**: Computer vision library for image/video processing, feature detection, object recognition.

5. **Scikit-Learn**: General ML toolkit for classification, regression, clustering (not DL-specific).

6. **CNTK**: Microsoft's deep learning library, strong for RNNs and LSTMs.

7. **CNTK-Python**: Python wrapper for the CNTK C++ core.

8. **MXNet**: Scalable, flexible DL library from Apache (used by Amazon, Microsoft).

9. **Brain**: Intel's DL library for neural nets and NLP.

10. **CNTK-GPU**: GPU-accelerated version of CNTK.

**When choosing a library, consider:**
\* Ease of use (beginners: Keras, PyTorch).
\* Performance (large datasets/models: TensorFlow, CNTK).
\* Specific use case (CV: OpenCV; NLP: NLTK, spaCy). | Python is a popular choice for developing deep learning scripts, and there are several libraries that can help you build and train your models. Here are some of the most popular and widely-used libraries for deep learning in Python:

1. **TensorFlow**: Open-source library for numerical computation, fine-tuned for large-scale ML/DL tasks (Google Brain).
**Pros:** Large community, widely used, scalable, flexible.
**Cons:** Steeper learning curve, complex architecture.

2. **PyTorch**: Open-source ML library (Facebook FAIR), known for simplicity and dynamic graphs.
**Pros:** Easy to learn, flexible, great for rapid prototyping.
**Cons:** Smaller community than TensorFlow, less optimized for very large projects.

3. **Keras**: High-level neural nets API in Python (runs on TensorFlow, CNTK, Theano).
**Pros:** User-friendly, high-level, multi-backend.
**Cons:** Limited low-level control, can be slower at scale.

4. **CNTK**: Microsoft's Computational Network Toolkit.
**Pros:** High-performance, scalable, distributed training.
**Cons:** Smaller community, less widespread.

5. **Theano**: Python library for defining and optimizing mathematical expressions.
**Pros:** Fast, flexible, research-friendly.
**Cons:** Steep learning curve, less user-friendly.

6. **MXNet**: Apache's scalable DL library, multi-language support.
**Pros:** Lightweight, high-performance, scalable.
**Cons:** Smaller community, less mature.

7. **Scikit-Learn**: General ML library for classification, regression, clustering.
**Pros:** Simple, excellent docs, broad algorithm set.
**Cons:** Not DL-focused, less powerful for deep networks.

**When choosing a library, consider:**
\* Your experience level (beginners: Keras, TensorFlow).
\* Project scale (large: TensorFlow, PyTorch).
\* Project type (research: Theano, CNTK).
\* Level of control required (TensorFlow, PyTorch).

Ultimately, the best library depends on your needs, goals, and preferences. |

Table 13: The response generated by SimPO+ComPO outlines a clearer setup of the variables and how each step is formulated compared to that generated by SimPO.

**Question:** *Navina has $30 more to her weekly budget than her younger sister and can afford to get one of the many online credit cards she likes. What do they each have to spend?*

| Gemma-9B-Instruct-SimPO | Gemma-9B-Instruct-SimPO-ComPO |
|---|---|
| | Here's how to solve this problem: |
| | **Let's use variables:** |
| | * Let 'x' represent the amount Navina's sister has in her weekly budget.
* Navina has $30 more than her sister, so Navina has 'x + $30' in her budget. |
| Here's how to solve this problem: | **Answer:** |
| **Let's use variables:** | * *Sister's weekly budget:* x |
| * *Sister's weekly budget:* x
* *Navina's weekly budget:* x + $30 | * *Navina's weekly budget:* x + $30 |
| **Since we don't have a specific total budget to divide, we can only express their individual amounts:** | We need a specific number for 'x' to get exact amounts, but this setup shows the relationship between their budgets. |
| * *Sister:* x dollars per week
* *Navina:* x + $30 dollars per week | |

