# OpenReview forum: "ComPO: Preference Alignment via Comparison Oracles"
_NeurIPS.cc/2025/Conference — NeurIPS 2025 poster_

### Official Review · Reviewer_ArR9 · 2025-07-01

**Clarity:** 3
**Significance:** 2
**Originality:** 3
**Rating:** 4
**Confidence:** 3

**Summary:**

This paper introduces ComPO, a novel method for preference alignment based on comparison oracles. Motivated by the limitations of existing direct preference optimization (DPO) methods, specifically verbosity and likelihood displacement, the authors propose a zeroth-order optimization approach that treats pairwise human preferences as comparison oracle outputs.

**Questions:**

1. What is the specific version of the Mistral and Mistral-Instruct models? There are versions from v0.1 to v0.3. But in the link you put in the Appendix, it shows Zephyr, instead of Mistral model.

**Ethical Concerns:**

["NO or VERY MINOR ethics concerns only"]

**Final Justification:**

I decide to keep my score.

**Limitations:**

Yes.

**Paper Formatting Concerns:**

No.

**Quality:**

3

**Strengths And Weaknesses:**

Pros:
1. The paper addresses an important problem in alignment, i.e., the preference pairs in the dataset may be noisy.\
2. The method gives a theoretical guarantee on the resulting algorithm.\
3. The experiments are conducted on multiple models, which well support the claims of the paper.

Cons:
1. The algorithm introduces additional hyperparameters and computation, which may be hard to tune and adjust the trade-off on larger models and scale.\
2. There are previous works that also study similar problems in DPO, such as likelihood displacement, by simply conditioning on the reward scores [1, 2]. Intuitively, these approaches can also deal with the noisy preference data, since these pairs have similar reward scores and the algorithm does not struggle with disentangling them.\
3. Only the DPO baselines are compared in experiments. I would like to see the authors add discussions and comparisons with more related works, such as the above ones that use conditional DPO.


[1] Kim et al. "Margin matching preference optimization: Enhanced model alignment with granular feedback."\
[2] Zhang et al. "Reward-Augmented Data Enhances Direct Preference Alignment of LLMs."

---

> ### Author Rebuttal · Authors · 2025-07-25
>
> Thank you for your encouraging comments and positive evaluation! We reply to your main questions point-by-point below and have included these discussions in the revised version of our paper.
>
> 1. **The algorithm introduces additional hyperparameters and computation, which may be hard to tune and adjust the trade-off on larger models and scale.**
>
> We agree on the importance on training scalability. In respond to your concerns, we have conducted experiments to investigate the effect of number of perturbations, number of perturbed layers, gradient entry threshold and number of noisy pairs used.
>
> To study the effect of number of perturbations $m$, we conducted experiments on Mistral-7B-Instruct using 5 independent trials for each setting (all other hyperparameters unchanged). The results are:
>
> $$
> \\begin{array}{c|cc}
> \\textbf{Perturbation ($m$)} & \\textbf{WR \\% - mean}\\,\\pm\\,\\textbf{std (best)} & \\textbf{LC \\% - mean}\\,\\pm\\,\\textbf{std (best)}\\\\ \\hline
> 800  & 17.32\\,\\pm\\,0.86\\;(17.94) & 24.72\\,\\pm\\,1.02\\;(25.12)\\\\
> 1600 & 17.50\\,\\pm\\,0.65\\;(18.32) & 25.02\\,\\pm\\,0.91\\;(26.17)\\\\
> 3300 & 19.21\\,\\pm\\,0.58\\;(20.25) & 25.91\\,\\pm\\,0.95\\;(27.14)\\\\
> 5400 & 19.69\\,\\pm\\,0.36\\;(20.07) & 26.49\\,\\pm\\,0.81\\;(27.20)\\\\
> \\end{array}
> $$
>
> We observe that increasing $m$ improves both WR and LC, at the cost of higher computation time. Importantly, peak memory usage remains unchanged, as we do not store all perturbation vectors $\\{y_i\\}_{1 \leq i \leq m}$ -- only the running average gradient estimate is maintained (see Line 5 of Algorithm 2).
>
> We conduct experiments to perturb 3 layers (MLPs in layers 30–31 and the output layer) in Mistral-7B-Instruct, keeping all other parameters the same. The results are:
>
> $$\\begin{array}{c|cc|c}
> \\textbf{Layer perturbed (parameters)} & \\textbf{AE-WR \\% - mean}\\,\\pm\\,\\textbf{std (best)} & \\textbf{AE-LC \\% - mean}\\,\\pm\\,\\textbf{std (best)}\\ & \\textbf{AH (GPT-4.1)-WR  \\% - mean}\\,\\pm\\,\\textbf{std (best)}\\\\ \\hline
> 1 (0.13B)  & 17.50\\,\\pm\\,0.65\\;(18.32) & 25.02\\,\\pm\\,0.91\\;(26.17) & 10.80\\,\\pm\\,0.21 \\;(11.0)\\\\
> 3 (0.25B) & 18.19\\,\\pm\\,0.81\\;(19.38)  & 26.00\\,\\pm\\,0.89\\;(27.09)  & 11.26\\,\\pm\\,0.36 \\;(11.7)\\\\
> \\end{array}$$
>
> Notably, we adopt the most recent version of Arena‑Hard, which uses GPT‑4.1 as the judge -- an upgrade from GPT‑4 Turbo used in our original results. This change significantly impacts evaluation outcomes: for example, DPO’s WR drops from 14.4% (under GPT‑4 Turbo) to 10.5% (under GPT‑4.1), whereas ComPO's results show consistent improvement. This highlights the robustness of our approach under stronger evaluation settings.
>
> Perturbing more layers leads to better performance due to a richer gradient direction set. The GPU memory increases from 16.3GB to 16.7GB, and processing 600 perturbations takes 60 seconds (vs. 50 seconds), which we consider reasonable.
>
> We conducted ablation studies on the gradient entry threshold $\lambda_g$ -- which selectively updates only high-magnitude gradient entries -- under $m=3300$ with 5 independent trials:
>
> $$
> \\begin{array}{c|c|cc}
> \\textbf{Threshold $\\lambda_{g}$} & \\textbf{\\% updated gradient entry} & \\textbf{WR \\% – mean}\\,\\pm\\,\\textbf{std (best)} & \\textbf{LC \\% – mean}\\,\\pm\\,\\textbf{std (best)}\\\\
> \\hline
> 0       & 100\\%  & 15.72\\,\\pm\\,0.77\\;(16.34) & 23.42\\,\\pm\\,1.03\\;(24.28)\\\\
> 0.00004 &  63\\%  & 16.02\\,\\pm\\,0.69\\;(16.69) & 24.01\\,\\pm\\,0.91\\;(25.10)\\\\
> 0.00018 &   6\\%  & 19.02\\,\\pm\\,0.62\\;(20.15) & 26.06\\,\\pm\\,0.81\\;(27.27)\\\\
> 0.00022 &   1\\%  & 19.21\\,\\pm\\,0.58\\;(20.25) & 25.91\\,\\pm\\,0.95\\;(27.14)\\\\
> 0.00025 &  0.15\\% & 16.10\\,\\pm\\,0.11\\;(16.21) & 23.82\\,\\pm\\,0.23\\;(24.00)\\\\
> \\end{array}
> $$
>
> We found that setting $\lambda_g$ to retain 1\%-5\% of entries offers the best trade-off between performance and stability. Too high or too low thresholds degrade performance.
>
> In the submitted paper, we reported ComPO’s performance using only the first 100 noisy preference pairs. Here, we extend the analysis to 300 noisy pairs and observe consistent improvements on both AlpacaEval2 (AE) and Arena‑Hard (AH).
>
> $$
> \\begin{array}{c|cc|c}
> \\textbf{\\# of noisy pairs} & \\textbf{AE-WR \\% - mean}\\,\\pm\\,\\textbf{std (best)} & \\textbf{AE-LC \\% - mean}\\,\\pm\\,\\textbf{std (best)}\\ & \\textbf{AH (GPT-4.1)-WR  \\% - mean}\\,\\pm\\,\\textbf{std (best)}\\\\ \\hline
> 100  &19.21\\,\\pm\\,0.58\\;(20.25) & 25.91\\,\\pm\\,0.95\\;(27.14) & 11.02\\,\\pm\\,0.13\\;(11.2) \\\\
> 300 & 20.07\\,\\pm\\,0.99\\;(21.35) & 26.28\\,\\pm\\,0.81\\;(27.59) & 11.76\\,\\pm\\,0.30\\;(12.1) \\\\
> \\end{array}
> $$
>
> Due to time constraints, we used the same hyperparameters as in the 100-sample training without additional tuning. However, we believe performance could be further improved with targeted tuning, particularly of the learning rate. We have also publicly released the best-performing checkpoints to support follow-up research from the community.
>
> 2. **There are previous works that also study similar problems in DPO, such as likelihood displacement, by simply conditioning on the reward scores [1, 2]. Intuitively, these approaches can also deal with the noisy preference data, since these pairs have similar reward scores and the algorithm does not struggle with disentangling them. Only the DPO baselines are compared in experiments. I would like to see the authors add discussions and comparisons with more related works, such as the above ones that use conditional DPO.**
>
> We acknowledge that learning from noisy preference data has been studied in prior works, including those leveraging reward scores via conditional DPO [1, 2], and we will cite and discuss them in the revised manuscript. That said, our contribution lies in how we address noisy preferences. ComPO introduces a principled method for incorporating specialized comparison oracles. This approach not only offers theoretical guarantees but demonstrates strong empirical performance in large-scale experiments, setting it apart from existing methods.
>
> We appreciate your suggestion to include comparisons with conditional DPO. Notably, conditional DPO modifies the DPO objective by conditioning on reward scores and solving it via gradient-based methods. This formulation is compatible with ComPO and can be combined in a similar way as we did with SimPO+ComPO. We will discuss this connection and provide comparisons in the revised manuscript.
>
> 3. **What is the specific version of the Mistral and Mistral-Instruct models? There are versions from v0.1 to v0.3. But in the link you put in the Appendix, it shows Zephyr, instead of Mistral model.**
>
> Thank you for pointing this out. Zephyr is an instruction-tuned model built on top of the Mistral-7B-v0.1 base model, effectively serving as a Mistral-Instruct SFT variant. We will clarify this and explicitly state the version used in the final version of the paper.
>
> We thank you again for your detailed reading and your constructive input! We hope and trust that our replies have alleviated your concerns, and we look forward to an open-minded discussion if any such concerns remain.

---

> > ### Comment · Reviewer_ArR9 · 2025-08-03
> > **Reponse to authors**
> >
> > Thanks for the detailed response. I decide to maintain my positive score.

---

> ### Author Response · Authors · 2025-08-04
> **Thank you!**
>
> Dear Reviewer ArR9,
>
> We really appreciate your detailed and constructive review, which is extremely helpful for us to conduct additional hyperparameter scaling experiment.
>
> In the end, thank you for your support to our work!
>
> ---
>
> Sincerely,
>
> Authors of Paper 202

---

### Official Review · Reviewer_2jaZ · 2025-07-02

**Clarity:** 3
**Significance:** 3
**Originality:** 3
**Rating:** 4
**Confidence:** 3

**Summary:**

This paper introduces a new preference alignment algorithm, ComPO, which uses comparison oracles and shows how this method can be easily unified with other preference alignment algorithms like DPO and SimPO.

**Questions:**

1. This method trains only a subset of the total parameters. Could there be further performance improvements if all parameters were trained?
2. Could this method improve performance on other tasks such as mathematics or coding, beyond preference alignment?

**Ethical Concerns:**

["NO or VERY MINOR ethics concerns only"]

**Final Justification:**

The reviewer's response has addressed most of my questions, and I have accordingly raised the score

**Limitations:**

Yes

**Quality:**

2

**Strengths And Weaknesses:**

strengths
1. This paper conducts experiments on various models and preference algorithms.
2. The suggested method is mathematically proven and presents a novel approach compared to existing methods.
3. This research is easily compatible with major alignment algorithms such as DPO and SimPO

Weaknesses
1. The baseline performance is strange. For example, in Table 1, the Llama-3-Instruct-8B with the UltraFeedback dataset setup is very similar to the SimPO paper's experimental setup. However, SimPO achieves 40.3 on AlpacaEval 2.0 LC and 32.6 on Arena Hard based on DPO, which differs significantly from the reported scores of 32.59 and 22.9 in this paper.
2. This paper does not display the pre-training (before preference alignment) model performance in the table. According to the SimPO paper, the Llama-3-Instruct-8B's Arena Hard performance is 22.3, which means DPO training shows virtually no performance improvement. This is just one example, and the main table should include both pre- and post training performance for all experimental results, showing relative performance improvements in tabular form. While it is understandable that training setups may vary across different experimental environments, the discrepancy here is too severe.
3. The memory efficiency advantage of ComPO claimed in the paper is limited. ComPO requires training DPO or SimPO on clean data first, followed by ComPO training, which ultimately means that memory-intensive training operations must still be performed.

---

> ### Author Rebuttal · Authors · 2025-07-25
>
> Thank you for your time and your input. We hope that our answers below will convince you about the merits of our work. We answer your questions below one-by-one, and have included these discussions in a revised version of our paper.
>
> 1. **The baseline performance is strange. In Table 1, the Llama-3-Instruct-8B with the UltraFeedback dataset setup is very similar to the SimPO paper's experimental setup. However, SimPO achieves 40.3 on AlpacaEval-2.0 LC and 32.6 on Arena-Hard based on DPO, which differs significantly from the reported scores of 32.59 and 22.9 in this paper.**
>
> Thank you for pointing this out. The discrepancy largely stems from differences in evaluation versions. Specifically, the SimPO paper reported results using an outdated version of AlpacaEval2 (v0.6.2), which has since been revised due to changes in vLLM decoding. This issue is acknowledged by the SimPO authors themselves on their GitHub:
>
> >*“AlpacaEval has a major revision for vllm decoding since 0.6.3 and causes a discrepancy from our experiments.”*
>
> In contrast, our experiments use AlpacaEval2 v0.6.6 (explicitly noted on line 758 of our paper), which provides more accurate and consistent evaluations. For example, we re-evaluated SimPO's Llama-3-Instruct-8B-SimPO-v0.2 checkpoint using v0.6.6 and observed an LC score of 49.53—lower than the 53.7 reported on their github page, further confirming the evaluation shift.
>
> Regarding training: to perform ablations on clean vs. noisy preference pairs, we train DPO (for Mistral-7B and Llama-3-8B) on clean data using H100 GPUs, use SimPO checkpoint (for Gemma-2-9B) as released without retraining. For consistency and reproducibility, we followed default TRL (huggingface/trl from Github) hyperparameters. Notably, our SimPO+ComPO results do use the official SimPO checkpoint for fair comparison.
>
> 2. **This paper does not display the pre-training (before preference alignment) model performance in the table. According to the SimPO paper, the Llama-3-Instruct-8B's Arena Hard performance is 22.3, which means DPO training shows virtually no performance improvement. This is just one example, and the main table should include both pre- and post training performance for all experimental results, showing relative performance improvements in tabular form...**
>
> We agree that including pre-training/SFT baselines is important for evaluating relative gains. We will add all relevant SFT (preference-untrained) performance evaluation results in the revised version.
>
> Regarding the specific Llama-3-Instruct-SFT checkpoint: the one used in SimPO is no longer available on Hugging Face. As such, we used Meta’s official Llama-3-Instruct release, referenced explicitly in our appendix (page 20). We note that this might not be a perfect match to SimPO’s original setup. In addition, Arena-Hard has since been updated to use GPT-4.1 as the judge recently, while our results in the submitted paper rely on GPT-4-turbo. We have summarized supplementary evaluations using the updated judge (see our answers to Q4), which shows that ComPO achieves a consistent improvement.
>
> 3. **The memory efficiency advantage of ComPO claimed in the paper is limited. ComPO requires training DPO or SimPO on clean data first, followed by ComPO training, which ultimately means that memory-intensive training operations must still be performed.**
>
> Thank you for highlighting this. We clarify that our two-stage training (DPO + ComPO) in Table 1 was designed for ablation purposes: to isolate the effect of noisy vs. clean pairs. It demonstrates DPO’s limitations with noisy data, as evidenced by the minimal performance difference between DPO and DPO\_clean.
>
> In practical deployment, one may conduct the alignment process as analogous to the pre-training vs. post-training paradigm. A user may either choose to run both DPO/SimPO and ComPO together or, more likely, choose not to perform the initial, memory-heavy DPO/SimPO training themselves. Instead, they would start with an existing, publicly available model that has already been aligned on a general, high-quality dataset (e.g., DPO-tuned checkpoints). ComPO empowers a user to take that pre-aligned public model and further refine it using their own, potentially noisy and task-specific, preference data, using  a more affordable GPU platform.
>
> As shown in Table 2, applying ComPO directly on existing, well-tuned SimPO checkpoints allows ComPO to extract value from noisy data without retraining from scratch, offering a more memory-efficient and scalable approach in real-world settings.
>
> 4. **This method tunes a subset of the total parameters. Could there be further performance improvements if all parameters were trained?**
>
> We agree and have conducted experiments to investigate the effect of number of perturbations, number of perturbed layers, gradient entry threshold and number of noisy pairs used.
>
> To study the effect of number of perturbations $m$, we conducted experiments on Mistral-7B-Instruct using 5 independent trials for each setting (all other hyperparameters unchanged). The results are:
>
> $$\\begin{array}{c|cc}
> \\textbf{Perturbation ($m$)} & \\textbf{WR \\% - mean}\\,\\pm\\,\\textbf{std (best)} & \\textbf{LC \\% - mean}\\,\\pm\\,\\textbf{std (best)}\\\\ \\hline
> 800  & 17.32\\,\\pm\\,0.86\\;(17.94) & 24.72\\,\\pm\\,1.02\\;(25.12)\\\\
> 1600 & 17.50\\,\\pm\\,0.65\\;(18.32) & 25.02\\,\\pm\\,0.91\\;(26.17)\\\\
> 3300 & 19.21\\,\\pm\\,0.58\\;(20.25) & 25.91\\,\\pm\\,0.95\\;(27.14)\\\\
> 5400 & 19.69\\,\\pm\\,0.36\\;(20.07) & 26.49\\,\\pm\\,0.81\\;(27.20)\\\\
> \\end{array}$$
>
> We observe that increasing $m$ improves both WR and LC, at the cost of higher computation time. Importantly, peak memory usage remains unchanged, as we do not store all perturbation vectors $\\{y_i\\}_{1 \leq i \leq m}$ -- only the running average gradient estimate is maintained (see Line 5 of Algorithm 2).
>
> We conduct experiments to perturb 3 layers (MLPs in layers 30–31 and the output layer) in Mistral-7B-Instruct, keeping all other parameters the same. The AlpacaEval2 (AE) and ArenaHard (AH) results are:
>
> $$\\begin{array}{c|cc|c}
> \\textbf{Layer perturbed (parameters)} & \\textbf{AE-WR \\% - mean}\\,\\pm\\,\\textbf{std (best)} & \\textbf{AE-LC \\% - mean}\\,\\pm\\,\\textbf{std (best)}\\ & \\textbf{AH (GPT-4.1)-WR  \\% - mean}\\,\\pm\\,\\textbf{std (best)}\\\\ \\hline
> 1 (0.13B)  & 17.50\\,\\pm\\,0.65\\;(18.32) & 25.02\\,\\pm\\,0.91\\;(26.17) & 10.80\\,\\pm\\,0.21 \\;(11.0)\\\\
> 3 (0.25B) & 18.19\\,\\pm\\,0.81\\;(19.38)  & 26.00\\,\\pm\\,0.89\\;(27.09)  & 11.26\\,\\pm\\,0.36 \\;(11.7)\\\\
> \\end{array}$$
>
> We adopt the most recent version of Arena‑Hard with GPT‑4.1 as the judge -- an upgrade from GPT‑4 Turbo used in our original results. This change significantly impacts evaluation outcomes: for example, DPO’s WR drops from 14.4\% (under GPT‑4 Turbo) to 10.5\% (under GPT‑4.1), while ComPO's results show consistent improvement under stronger evaluation settings.
>
> Perturbing more layers leads to better performance due to a richer gradient direction set. The GPU memory increases from 16.3GB to 16.7GB, and processing 600 perturbations takes 60 seconds (vs. 50 seconds), which we consider reasonable.
>
> We conducted ablation studies on the gradient entry threshold $\lambda_g$ -- which selectively updates only high-magnitude gradient entries -- under $m=3300$ with 5 independent trials:
>
> $$\\begin{array}{c|c|cc}
> \\textbf{Threshold $\\lambda_{g}$} & \\textbf{\\% updated gradient entry} & \\textbf{WR \\% – mean}\\,\\pm\\,\\textbf{std (best)} & \\textbf{LC \\% – mean}\\,\\pm\\,\\textbf{std (best)}\\\\
> \\hline
> 0  & 100\\% & 15.72\\,\\pm\\,0.77\\;(16.34) & 23.42\\,\\pm\\,1.03\\;(24.28)\\\\
> 0.00004 &  63\\% & 16.02\\,\\pm\\,0.69\\;(16.69) & 24.01\\,\\pm\\,0.91\\;(25.10)\\\\
> 0.00018 &  6\\% & 19.02\\,\\pm\\,0.62\\;(20.15) & 26.06\\,\\pm\\,0.81\\;(27.27)\\\\
> 0.00022 & 1\\% & 19.21\\,\\pm\\,0.58\\;(20.25) & 25.91\\,\\pm\\,0.95\\\;(27.14)\\\\
> 0.00025 & 0.15\\% & 16.10\\,\\pm\\,0.11\\;(16.21) & 23.82\\,\\pm\\,0.23\\;(24.00)\\\\
> \\end{array}$$
>
> We found that setting $\lambda_g$ to retain 1\%-5\% of entries offers the best trade-off between performance and stability. Too high or too low thresholds degrade performance.
>
> In the submitted paper, we reported ComPO’s performance using only 100 noisy preference pairs. Here, we extend the analysis to 300 noisy pairs and observe consistent improvements on both AE and AH:
>
> $$
> \\begin{array}{c|cc|c}
> \\textbf{\\# of noisy pairs} & \\textbf{AE-WR \\% - mean}\\,\\pm\\,\\textbf{std (best)} & \\textbf{AE-LC \\% - mean}\\,\\pm\\,\\textbf{std (best)}\\ & \\textbf{AH (GPT-4.1)-WR  \\% - mean}\\,\\pm\\,\\textbf{std (best)}\\\\ \\hline
> 100  &19.21\\,\\pm\\,0.58\\;(20.25) & 25.91\\,\\pm\\,0.95\\;(27.14) & 11.02\\,\\pm\\,0.13\\;(11.2) \\\\
> 300 & 20.07\\,\\pm\\,0.99\\;(21.35) & 26.28\\,\\pm\\,0.81\\;(27.59) & 11.76\\,\\pm\\,0.30\\;(12.1) \\\\
> \\end{array}
> $$
>
> Due to time constraints, we used the same hyperparameters as in the 100-sample training. However, we believe performance could be further improved with targeted tuning, particularly the learning rate.
>
> 5. **Could this method improve performance on other tasks such as mathematics or coding, beyond preference alignment?**
>
> Thank you for this thoughtful suggestion. While ComPO is designed to address issues like verbosity and likelihood displacement, it is less clear whether these issues are bottlenecks in tasks such as mathematics or coding. In those domains, the challenges often involve reasoning fidelity and structured problem solving, where methods like PPO or GRPO that directly optimize task-specific reward signals (e.g., via CoT) are typically more effective. We see this as a valuable direction for future work.
>
> We sincerely appreciate your close reading and constructive feedback. We hope our responses clarify the key points of our approach and address your concerns. Please don’t hesitate to reach out for further discussion or clarification.

---

> > ### Comment · Reviewer_2jaZ · 2025-08-06
> >
> > Thank you for your response. I acknowledge that the benchmark criteria have changed compared to the SIMPO paper. However, separately from that issue, the scores of the non-fine-tuned model should be disclosed, and without knowing these values precisely, I still find it difficult to make a proper judgment. Furthermore, the requirement for a pre-alignment trained model remains unchanged, and given that this approach does not replace existing DPO training, I find it difficult to accept memory efficiency as a advantage. Therefore, I will maintain my score.

---

> ### Author Response · Authors · 2025-08-06
> **Thank you for your comment! We provide full table and explanation to memory efficiency**
>
> Dear Reviewer 2jaZ,
>
> Thank you for your time and continued questions! We especially appreciate you for explicitly listing out your remaining concerns.
>
> ---
>
> 1. **the scores of the non-fine-tuned model should be disclosed, and without knowing these values precisely, I still find it difficult to make a proper judgment.**
>
> We clarify that due to the limit of rebuttal length, we cannot present all four tables. Therefore, we have already updated it in our revised version: "pre" indicates pre-alignment model without DPO finetuning (AE for AlpacaEval2, AH for Arena-Hard, MT for MT-Bench, all benchmarks are corresponding version presented in the original paper):
> $$\\begin{array}{c|cc|c|ccc} \\textbf{Llama-8B-Base}&\\textbf{AE-LC}&\\textbf{AE-WR} & \\textbf{AH-WR} & \\textbf{MT-Turn 1} & \\textbf{MT-Turn 2} & \\textbf{MT-avg}\\\\ \\hline
>  {\\text{PRE}}&3.21&7.97&4.1&6.53&5.66&6.10\\\\
> \\text{DPO}&4.14&10.43&12.1&6.61&5.85&6.23\\\\
> \\text{DPO-clean}&4.28&9.81&12.0&6.64&6.01&6.33\\\\
> \\text{ComPO}&5.39&10.93&12.1&6.60&6.28&6.44\\\\\\hline \\end{array}$$
> $$\\begin{array}{c|cc|c|ccc}\\textbf{Llama-8B-Instruct}&\\textbf{AE-LC}&\\textbf{AE-WR}&\\textbf{AH-WR}&\\textbf{MT-Turn 1}&\\textbf{MT-Turn 2}&\\textbf{MT-avg}\\\\\\hline
>  {\\text{PRE}}&24.06&23.69&20.8&8.22&7.57&7.90\\\\
> \\text{DPO}&32.59&31.99&22.9&8.30&7.55&7.93\\\\
> \\text{DPO-clean}&32.92&32.42&22.9&8.26&7.63&7.94\\\\
> \\text{ComPO}&35.79&35.03&23.1&8.39&7.71&8.05\\\\\\hline \\end{array}$$
> $$\\begin{array}{c|cc|c|ccc}\\textbf{Mistral-7B-Instruct}&\\textbf{AE-LC}&\\textbf{AE-WR}&\\textbf{AH-WR}&\\textbf{MT-Turn 1}&\\textbf{MT-Turn 2}&\\textbf{MT-avg}\\\\\\hline
>  {\\text{PRE}}&16.54&12.43&10.9&6.19&5.10&5.65\\\\
> \\text{DPO}&24.14&16.71&14.4&6.28&5.42&5.86\\\\
> \\text{DPO-clean}&23.89&16.15&14.2&6.11&5.34&5.73\\\\
> \\text{ComPO}&26.17&18.32&10.5&7.78&7.63&7.69\\\\\\hline \\end{array}$$
> $$\\begin{array}{c|cc|c|ccc} \\textbf{Mistral-7B-Base}&\\textbf{AE-LC}&\\textbf{AE-WR}&\\textbf{AH-WR}&\\textbf{MT-Turn 1}&\\textbf{MT-Turn 2}&\\textbf{MT-avg}\\\\\\hline
>  {\\text{PRE}}&7.33&4.48&1.1&6.10&5.04&5.57\\\\
> \\text{DPO}&9.71&6.27&2.9&6.20&5.38&5.79\\\\
> \\text{DPO-clean}&9.41&6.52&3.0&6.18&5.22&5.70\\\\
> \\text{ComPO}&11.66&6.55&3.2&6.22&5.32&5.77\\\\ \\hline \\end{array}$$
> All the results above are the original results from Table 1 without scalability experiment on ComPO. The results with more layers, noisy samples, and perturbations are presented under the Q4 our original rebuttal.
>
> ---
>
> 2. **The requirement for a pre-alignment trained model remains unchanged, and given that this approach does not replace existing DPO training, I find it difficult to accept memory efficiency as a advantage.**
>
> Thank you for your comment, and we apologize for any confusion caused. We would like to clarify that the key advantage of ComPO lies in its ability to improve performance without increasing peak memory usage. We will make this point clearer in the revised manuscript.
>
> Regarding the need for a pre-aligned model, we acknowledge that this requirement remains unchanged. In practice, however, well-tuned checkpoints -- such as those produced by DPO on specific preference datasets -- are publicly available to the community. Users can therefore start from these publicly available checkpoints and further refine them with ComPO on checkpoint's noisy pairs.
>
> To directly illustrate this, we applied ComPO to DPO checkpoints **without** separating noisy and clean pairs and observed comparable improvements (with $m=3300$), which shows that ComPO can effectively exploit noisy pairs and allow users with an alternative choice to run it directly on existing checkpoints with its noisy pairs. **This is especially valuable if users don't have large-memory GPU but still want to finetune public checkpoints on the noisy pairs that are useful but cannot be effectively utilized by DPO.** We will make this explicitly clear in the revised manuscript.
>
> $$\\begin{array}{c|cc|c|ccc} \\textbf{Mistral-7B-Instruct}&\\textbf{AE-LC}&\\textbf{AE-WR}&\\textbf{AH(GPT4.1)-WR}&\\textbf{MT-Turn 1}& \\textbf{MT-Turn 2} & \\textbf{MT-avg}\\\ \\hline
>  {\\text{DPO}} &24.14&16.71&10.4&6.28&5.42&5.86\\\\
>  {\\text{DPO+ComPO}}&27.03&20.85&11.4&7.80&7.61&7.71\\\\\\hline
>  {\\text{DPO-clean}}&23.89&16.15&10.5&6.11&5.34&5.73\\\\
> \\text{DPO-clean+ComPO} &27.14&20.25&11.2&7.82&7.59&7.71\\\\\\hline \\end{array}$$
>
> Thank you again for your detailed and insightful review, for raising these valuable points, and for your continued engagement, which gives us the opportunity to provide further clarification.
>
> ---
>
> Sincerely,
>
> Authors of Paper 202

---

> > ### Comment · Reviewer_2jaZ · 2025-08-07
> >
> > Thank you for the detailed response. My concerns have been addressed, and accordingly, I will raise the score

---

> > > ### Author Response · Authors · 2025-08-07
> > > **Thank you for your time and effort!**
> > >
> > > Dear Reviewer 2jaZ,
> > >
> > > We sincerely appreciate for your time and effort in providing insightful and detailed review. We will incorporate the discussions into the revised version.
> > >
> > > We again thank you for your comments and suggestions, which is important for us to improve ComPO.
> > >
> > > ---
> > >
> > > Sincerely,
> > >
> > > Authors of Paper 202

---

### Official Review · Reviewer_JNR3 · 2025-07-03

**Clarity:** 3
**Significance:** 2
**Originality:** 3
**Rating:** 4
**Confidence:** 4

**Summary:**

This paper focuses on the verbosity and likelihood displacement problems in direct preference alignment. The authors assumed these challenges are caused by noisy pairwise preference data which induces similar likelihood of preferred and dispreferred responses, and then propose ComPO, a zeroth-order method, which directly approximates and optimizes the model parameters according to the preference comparison oracle (whether the current parameters can prefer a chosen response compared to the rejected one). To avoid unaffordable computational cost, the authors further simply the framework as a lightweight component by only updating the output layer. Experiments show that ComPO can achieve better results compared to the original DPO or SimPO with acceptable costs.

**Questions:**

1. Can you further explain why your method can address the verbosity issue?

2. In Sec.4.2, especially Fig.1 (b) and (c), can you provide the memory usage and run time of i) the original DPO, and ii) several previous methods?

**Ethical Concerns:**

["NO or VERY MINOR ethics concerns only"]

**Final Justification:**

I read the authors' response, and confirm that part of my concerns are addressed.

**Limitations:**

Yes.

**Quality:**

3

**Strengths And Weaknesses:**

**[Strengths]**
1. This paper is well-motivated and clearly written, and focus on likelihood displacement, an important challenge in the alignment community.
2. The proposed method, ComPO, and the idea of using comparison oracles and zeroth-order approximation is novel and insightful.
3. The authors provided rigorous proof of ComPO’s convergence and also developed a practical and efficient algorithm.
4. The author conducted extensive experiments on different backbone LLMs and alignment methods.

**[Summary Of Weaknesses]**

There are two concerns:
1. The challenge of verbosity has not been sufficiently analyzed or explained. It’s unclear why ComPO  can address the verbosity problem. The whole method, algorithm, and theoretical discussions seem only connected to likelihood displacement.

2. The biggest problem lies in the experiment settings. In detail:

a. The authors didn’t compare ComPO with any existing relevant methods, but only the original DPO/SimPO, as well as a variant of ComPO, i.e., ComPO_clean, which is not the original one used in (Razin et al., ICLR 2025), as the authors didn’t use CHES score. Besides, none of existing methods for mitigation displacement is compared, e.g., (Amini et al., 2024) or (Xiao et al., 2024).

b. The comparison in Tables 1& 2 is unfair. The authors report the best results of ComPO among five runs, which fails to reflect the real average performance of ComPO. If we check Table 4 in Appendix, where the authors reported average performance over 5 runs, we can find the improvement is quite marginal and ComPO even performs worse. For example, under Arena-Hard, DPOclean+ComPO (Mistral-instruct-7B) is much worse than the original DPO (9.94 vs 14.4).

c. Even we consider the results in Table 1, the improvement is quite marginal. This might be because the authors only fine-tuned a small part of all parameters (1%). To further demonstrate the effectiveness of ComPO, the authors should conduct an ablation study on the proportion of tuned parameters.

Generally, I believe this is a novel and interesting paper. If the authors can address the concerns above, I will change my score.

[Reference]:

Xiao  et al., Cal-DPO: Calibrated Direct Preference Optimization for Language Model Alignment. NeurIPS 2024.

---

> ### Author Rebuttal · Authors · 2025-07-26
>
> Thank you for your time and your input. We answer your questions below one-by-one, and have included these discussions in a revised version of our paper.
>
> 1. **The comparison in Tables 1 and 2 is unfair. The authors report the best results of ComPO among 5 runs, which fails to reflect the real average performance of ComPO. If we check Table 4 in Appendix, where the authors reported average performance over 5 runs, we can find the improvement is quite marginal and ComPO performs worse. For example, under Arena-Hard, DPO\_clean+ComPO (Mistral-instruct-7B) is much worse than the original DPO (9.94 vs 14.4).**
>
> We understand the concern, but argue that the improvement on specific benchmark and model setup can be assessed by increasing the number of perturbations ($m$) and the number of noisy preference pairs that we allow ComPO to use.
>
> We used $m=1600$ in the submitted paper. We recently tested a larger value of $m$ with Mistral-7B-Instruct and AlpacaEval2 (with all other parameters being the same as in the submitted paper). They consist of 5 independent runs.
>
> $$
> \\begin{array}{c|cc}
> \\textbf{Perturbation ($m$)} & \\textbf{WR \\% - mean}\\,\\pm\\,\\textbf{std (best)} & \\textbf{LC \\% - mean}\\,\\pm\\,\\textbf{std (best)}\\\\ \\hline
> 800  & 17.32\\,\\pm\\,0.86\\;(17.94) & 24.72\\,\\pm\\,1.02\\;(25.12)\\\\
> 1600 & 17.50\\,\\pm\\,0.65\\;(18.32) & 25.02\\,\\pm\\,0.91\\;(26.17)\\\\
> 3300 & 19.21\\,\\pm\\,0.58\\;(20.25) & 25.91\\,\\pm\\,0.95\\;(27.14)\\\\
> 5400 & 19.69\\,\\pm\\,0.36\\;(20.07) & 26.49\\,\\pm\\,0.81\\;(27.20)\\\\
> \\end{array}
> $$
>
> Second, we reported ComPO’s performance based on training it with only 100 noisy preference pairs in the submitted paper. Here, we extend the analysis to 300 noisy pairs and observe consistent improvements on both AlpacaEval2 (AE) and Arena‑Hard (AH):
>
> $$
> \\begin{array}{c|cc|c}
> \\textbf{\\# of noisy pairs} & \\textbf{AE-WR \\% - mean}\\,\\pm\\,\\textbf{std (best)} & \\textbf{AE-LC \\% - mean}\\,\\pm\\,\\textbf{std (best)}\\ & \\textbf{AH (GPT-4.1)-WR  \\% - mean}\\,\\pm\\,\\textbf{std (best)}\\\\ \\hline
> 100  &19.21\\,\\pm\\,0.58\\;(20.25) & 25.91\\,\\pm\\,0.95\\;(27.14) & 11.02\\,\\pm\\,0.13\\;(11.2) \\\\
> 300 & 20.07\\,\\pm\\,0.99\\;(21.35) & 26.28\\,\\pm\\,0.81\\;(27.59) & 11.76\\,\\pm\\,0.30\\;(12.1) \\\\
> \\end{array}
> $$
>
> Note that we adopted the most recent version of Arena‑Hard that uses GPT‑4.1 as the judge, available only after we wrap up the initial submission. GPT-4.1 is a judge much stronger than GPT‑4 Turbo and significantly impacts evaluation outcomes: for example, DPO’s WR drops from 14.4 (under GPT‑4 Turbo) to 10.5 (under GPT‑4.1), while ComPO's results show consistent improvement under this stronger and more robust judge. This result highlights the robustness of our approach under stronger evaluation settings. Moreover, due to time constraints, we used the same hyperparameters as in the 100-sample training without additional tuning. With targeted tuning (e.g., to the learning rate), we believe the performance would further improve.
>
> 2. **Even we consider the results in Table 1, the improvement is quite marginal. This might be because the authors only fine-tuned a small part of all parameters (1\%). To further demonstrate the effectiveness of ComPO, the authors should conduct an ablation study on the proportion of tuned parameters.**
>
> We agree that exploring full-model or larger-parameter variants is important. We extended our experiments to perturb 3 layers (MLPs in layers 30–31 and the output layer) in Mistral-7B-Instruct, keeping all other parameters the same ($m=1600$). The results are:
>
> $$
> \\begin{array}{c|cc|c}
> \\textbf{Layer perturbed (parameters)} & \\textbf{AE-WR \\% - mean}\\,\\pm\\,\\textbf{std (best)} & \\textbf{AE-LC \\% - mean}\\,\\pm\\,\\textbf{std (best)}\\ & \\textbf{AH (GPT-4.1)-WR  \\% - mean}\\,\\pm\\,\\textbf{std (best)}\\\\ \\hline
> 1 (0.13B)  & 17.50\\,\\pm\\,0.65\\;(18.32) & 25.02\\,\\pm\\,0.91\\;(26.17) & 10.80\\,\\pm\\,0.21 \\;(11.0)\\\\
> 3 (0.25B) & 18.19\\,\\pm\\,0.81\\;(19.38)   & 26.00\\,\\pm\\,0.89\\;(27.09)  & 11.26\\,\\pm\\,0.36 \\;(11.7) \\\\
> \\end{array}
> $$
>
> Perturbing two more layers than the original 1 layer leads to better performance due to a richer gradient direction set. The GPU memory increases very mildly from 16.3GB to 16.7GB, and running time per 600 perturbations takes 60 seconds (vs. 50 seconds), which we consider reasonable.
>
> To ensure stability, ComPO uses a gradient entry threshold $\lambda_g$, which selectively updates only high-magnitude gradient entries. We conducted ablation studies under $m=3300$ with 5 independent trials:
>
> $$
> \\begin{array}{c|c|cc}
> \\textbf{Threshold $\\lambda_{g}$} & \\textbf{\\% updated gradient entry} & \\textbf{WR \\% – mean}\\,\\pm\\,\\textbf{std (best)} & \\textbf{LC \\% – mean}\\,\\pm\\,\\textbf{std (best)}\\\\
> \\hline
> 0       & 100\\%  & 15.72\\,\\pm\\,0.77\\;(16.34) & 23.42\\,\\pm\\,1.03\\;(24.28)\\\\
> 0.00004 &  63\\%  & 16.02\\,\\pm\\,0.69\\;(16.69) & 24.01\\,\\pm\\,0.91\\;(25.10)\\\\
> 0.00018 &   6\\%  & 19.02\\,\\pm\\,0.62\\;(20.15) & 26.06\\,\\pm\\,0.81\\;(27.27)\\\\
> 0.00022 &   1\\%  & 19.21\\,\\pm\\,0.58\\;(20.25) & 25.91\\,\\pm\\,0.95\\\;(27.14)\\\\
> 0.00025 &  0.15\\% & 16.10\\,\\pm\\,0.11\\;(16.21) & 23.82\\,\\pm\\,0.23\\;(24.00)\\\\
> \\end{array}
> $$
>
> We found that setting $\lambda_g$ to retain 1\%-5\% of entries offers the best trade-off between performance and stability. Too high or too low thresholds, understandably, degrade performance.
>
> 3. **The authors didn’t compare ComPO with any existing relevant methods, but only the original DPO/SimPO, as well as a variant of ComPO, i.e., ComPO_clean, which is not the original one used in (Razin et al., ICLR 2025), as the authors didn’t use CHES score. Besides, none of existing methods for mitigation displacement is compared, e.g., (Amini et al., 2024) or (Xiao et al., 2024).**
>
> We appreciate your suggestion to include comparisons with (Razin et al., ICLR 2025) and other existing methods, e.g., (Amini et al., 2024) or (Xiao et al., 2024). Notably, the CHES score (see Definition 2 in their paper) is calculated based on the hidden embedding of each token in the responses, thereby resulting in more computational efforts and bandwidth burden to manipulate large embedding vectors from a practical point of view.
>
> ComPO is designed to address the challenge of noisy preference labels—a common issue that can significantly degrade the performance of other preference optimization methods. From this perspective, ComPO is not intended as a direct replacement for existing methods, but rather as a complementary and modular component that enhances their robustness.
>
> We agree that comparing with methods like (Amini et al., 2024) and (Xiao et al., 2024) is important, and we appreciate the suggestion. These methods are perfect examples of the synergy we propose. Methods like (Xiao et al., 2024)—which adds MSE terms to calibrate log-likelihood ratios—tackle a different problem axis (reward value calibration) than ComPO (label noise robustness). Because these goals are orthogonal, our framework is directly compatible. ComPO can be integrated with them to make it more resilient to noisy data, much like we demonstrated with SimPO+ComPO.
>
> 4. **The challenge of verbosity has not been sufficiently analyzed or explained. It’s unclear why ComPO can address the verbosity problem. The whole method, algorithm, and theoretical discussions seem only connected to likelihood displacement.**
>
> We agree that verbosity is a nuanced and under-explored challenge in alignment. Recent work (e.g., Park et al., ACL 2024) shows that verbosity in DPO can be mitigated by incorporating appropriate regularization into the objective, suggesting that modifying the objective can better capture alignment goals.
>
> Although our ComPO does not use such a mod, ComPO's results consistently improve LC (length-controlled win rate), indicating that ComPO helps reduce verbosity. While our theoretical discussions focus only on likelihood displacement, we see verbosity as a valuable direction for future work. Perhaps, ComPO implicitly optimizes a more robust and alignment-faithful objective.
>
> 5. **In Sec.4.2, especially Fig.1 (b) and (c), can you provide the memory usage and run time of i) the original DPO, and ii) several previous methods?**
>
> Thank you for this excellent question. We've chosen not to directly compare wall-clock time and peak GPU memory across DPO, SimPO, and our ComPO due to the significant differences in their GPU hardware requirements. In our experiments, we trained DPO (for Mistral-7B and Llama-3-8B) on clean data on H100 GPUs. Using Gemma-2-9B-SimPO checkpoint as released, we trained ComPO models on much weaker A40 GPUs independently and in parallel. A direct numerical comparison of wall-clock time between a model trained on an H100 and one trained on an A40 would be fundamentally misleading, so we chose not to directly compare across devices in the paper.
>
> Let us report the memory usages and wall-clock times. The peak memory for running DPO and SimPO is 77GB and 69GB, respectively, on H100 GPUs for Llama-3-8B as the example, and running ComPO needed only around 23GB on A40 GPUs. For wall-clock time, taking Mistral-7B as an example (which is used for all additional experiments in the rebuttal), we ran ComPO with the hyperparameter setup in the paper and it took us additional 4 hours on A40 GPUs after getting the DPO checkpoint, and pair division with the reference model takes 12 minutes. We are committed to full transparency and will list detailed information for peak memory, wall-clock time, pair-division time across all models and setups in the revised version.
>
> We sincerely appreciate your close reading and constructive feedback. We hope our responses clarify the key points of our approach and address your concerns. Please don’t hesitate to reach out for further discussion or clarification -- we would welcome the opportunity to continue the dialogue.

---

> > ### Comment · Reviewer_JNR3 · 2025-08-06
> > **Thanks for your response**
> >
> > Thanks for the authors' responses and additional results. Part of my concerns have been addressed (the verbosity part and the marginal improvement issue), but the others remain (particularly, comparison with existing relevant methods. The authors claim ComPO can be combined with them, but they didn't provide empirical results to support the claimed compatibility). Consider the novelty and the theoretical contribution of this work, I decide to stand for this paper, and raised my score to 4. Please reframe your motivation and claim to reduce the emphasis on verbosity, as your method design is not grounded in it, and provide more comparison (or results on the combination with) existing methods.

---

> ### Author Response · Authors · 2025-08-06
> **Thank you for your suggestions, effort, and time!**
>
> Dear Reviewer JNR3,
>
> We sincerely appreciate your insightful review and continued engagement in sharing your suggestions. We fully agree on those suggestions, and in the revised manuscript, we will definitely reduce the emphasis on verbosity, and conduct further experiments to evidence the flexibility of our method.
>
> In the end, thank you for your support and valuable input towards ComPO.
>
> ---
>
> Sincerely,
>
> Authors of Paper 202

---

### Official Review · Reviewer_Hemu · 2025-07-04

**Clarity:** 3
**Significance:** 3
**Originality:** 4
**Rating:** 4
**Confidence:** 5

**Summary:**

The paper introduces ComPO, a novel method for preference alignment of large language models (LLMs) that directly leverages comparison oracles to address well-known issues in direct preference optimization (DPO), such as verbosity and likelihood displacement. The authors theoretically prove convergence guarantees for a basic scheme and further design a practical scheme that is efficient for large-scale models. Experiments on multiple models (Mistral-7B, Llama-3-8B, Gemma-2-9B) and benchmarks (AlpacaEval 2, MT-Bench, Arena-Hard) demonstrate that ComPO improves alignment by effectively utilizing noisy preference pairs that existing methods often discard or handle poorly.

**Questions:**

1. Could you clarify how sensitive ComPO’s performance is to the choice of threshold for separating noisy vs. clean preference pairs? Would a more adaptive or learned approach outperform a fixed threshold?
2. Could you report wall-clock time and peak GPU memory for DPO, SimPO, and your ComPO (or DPO + ComPO) pipeline （including the cost of using a reference model to divide the dataset）?
3. Your current scheme “hard-switches” objectives—DPO on clean pairs, ComPO on noisy ones. Have you explored a soft interpolation, weighting DPO and ComPO by a confidence score derived from the reference model?
4. Please also address concerns mentioned in the weaknesses.

**Ethical Concerns:**

["NO or VERY MINOR ethics concerns only"]

**Final Justification:**

The authors have provided detailed and thoughtful responses that satisfactorily address the main concerns I raised in my initial review. Specifically, the authors clarified the wall-clock time and GPU memory usage across models and setups, highlighting that ComPO is significantly more memory-efficient and practical for training on low-end GPUs. Their decision not to compare wall-clock times across DPO/SimPO/ComPO due to hardware heterogeneity is reasonable and well-explained. Also, new ablation results convincingly demonstrate how increasing the number of perturbations improves alignment performance while maintaining stable memory consumption. These experiments validate the scalability and robustness of ComPO’s gradient estimation.
Beyond output layer tuning: Additional experiments perturbing deeper layers show consistent performance gains and manageable computational overhead. The use of a gradient threshold to ensure training stability is well-motivated and backed by ablation results.
The authors have committed to restructuring Section 3, fixing broken references, and adding a high-level pipeline diagram and clearer metric definitions in the revised version, which will significantly improve readability.

While the overall alignment improvements remain incremental, the method is theoretically grounded, addresses meaningful challenges in preference optimization (e.g., verbosity, likelihood displacement), and is practically scalable. The authors’ rebuttal has strengthened my confidence in the work. I am satisfied with the authors’ feedback and will maintain my original score of accept.

**Limitations:**

YES

**Quality:**

3

**Strengths And Weaknesses:**

Strengths:
1 The use of comparison oracles for LLM preference alignment is creative and well-motivated, bridging zeroth-order optimization theory and practical LLM training.
2. The paper tackles two important practical challenges, verbosity and likelihood displacement, with clear analysis and empirical evidence.
3. The authors provide convergence guarantees under realistic non-convex settings, which strengthens the credibility of their method.
4.They design scalable approximations (e.g., output-layer perturbations, sparse gradient estimation) that make the approach feasible for billion-parameter LLMs.

Weaknesses:
1. Gradient estimation requires many perturbations; the paper reports memory usage but omits wall-clock **time** and sample counts, making practical cost hard to gauge. Reported win-rate improvements are typically 2-4 pp; The incremental may benefit insufficient relative to added training complexity.
2. Only the output projection layer is tuned; it is unclear whether full-model or larger-parameter variants would preserve gains or suffer instability.
3. The paper lacks a high-level pipeline diagram, does not define how Win-Rate (WR) and Length-Controlled Win-Rate (LC) are computed, and contains broken appendix references (e.g., line 272 cites Appendix B instead of E), all of which hinder clarity.
4. The paper does not analyze the impact of the number of perturbations (denoted as m) on performance. Since the comparison-oracle-based gradient estimation relies on multiple random perturbation directions, the choice of m directly affects the quality of the estimated gradient and thus the effectiveness of alignment.
5. The organization of the paper could be improved for better clarity. In particular, Section 3 is titled “Main Results,” but it mixes algorithmic design, theoretical analysis, and practical implementation details.

---

> ### Author Rebuttal · Authors · 2025-07-25
>
> Thank you for your encouraging comments and positive evaluation! We reply to your main questions point-by-point below and have included these discussions in the revised version of our paper.
>
> 1. **The gradient estimation requires many perturbations. While the paper reports memory usage, it omits wall-clock time and sample counts, making practical cost hard to gauge. The paper also does not report wall-clock time and peak GPU memory for DPO, SimPO, and your ComPO (or DPO + ComPO) pipeline (including the cost of using a reference model to divide the dataset)?**
>
> To clarify, in Figure 1 (right), we have reported the perturbation wall-clock running times of our ComPO on Mistral-7B, Llama-3-8B, and Gemma-2-9B. We've chosen not to directly compare wall-clock time and peak GPU memory across DPO, SimPO, and our ComPO. The core reason lies in the significant differences in their GPU hardware requirements. DPO and SimPO are typically trained on GPUs with larger memory, such as H100s, for large (7B, 8B) models (e.g., see arXiv:2405.14734), while our ComPO is remarkably memory efficient and can be trained on relatively low-end A40 GPUs. In our experiments, we trained DPO (for Mistral-7B and Llama-3-8B) on clean data on H100 GPUs. Using Gemma-2-9B-SimPO checkpoint as released, we trained ComPO models on A40 GPUs independently and in parallel. A direct numerical comparison of wall-clock time between a model trained on an H100 and one trained on an A40 would be fundamentally misleading, so we chose not to directly compare across devices in the paper.
>
> Let us report the memory usages and wall-clock times. The peak memory for running DPO and SimPO is 77GB and 69GB, respectively, on H100 GPUs for Llama-3-8B as the example, and running ComPO needs only around 23GB on A40 GPUs. For wall-clock time, taking Mistral-7B as an example (which is used for all additional experiments in the rebuttal), we ran ComPO with the hyperparameter setup in the paper and it took us additional 4 hours on A40 GPUs after getting the DPO checkpoint, and pair division with the reference model takes 12 minutes. We are committed to full transparency and will list detailed information for peak memory, wall-clock time, pair-division time across all models and setups in the revised version.
>
> 2. **The paper does not analyze the impact of the number of perturbations (denoted as $m$) on performance. Since the comparison-oracle-based gradient estimation relies on multiple random perturbation directions, the choice of $m$ directly affects the quality of the estimated gradient and thus the effectiveness of alignment.**
>
> Great question. To study the effect of $m$, we conducted new experiments on Mistral-7B-Instruct using 5 independent trials for each setting (all other hyperparameters are unchanged). The results are:
>
> $$
> \\begin{array}{c|cc}
> \\textbf{Perturbation ($m$)} & \\textbf{WR \\% - mean}\\,\\pm\\,\\textbf{std (best)} & \\textbf{LC \\% - mean}\\,\\pm\\,\\textbf{std (best)}\\\\ \\hline
> 800  & 17.32\\,\\pm\\,0.86\\;(17.94) & 24.72\\,\\pm\\,1.02\\;(25.12)\\\\
> 1600 & 17.50\\,\\pm\\,0.65\\;(18.32) & 25.02\\,\\pm\\,0.91\\;(26.17)\\\\
> 3300 & 19.21\\,\\pm\\,0.58\\;(20.25) & 25.91\\,\\pm\\,0.95\\;(27.14)\\\\
> 5400 & 19.69\\,\\pm\\,0.36\\;(20.07) & 26.49\\,\\pm\\,0.81\\;(27.20)\\\\
> \\end{array}
> $$
>
> Observe that increasing $m$ improves both Win-Rate (WR) and Length-Controlled Win-Rate (LC), at the cost of computation time. Importantly, peak memory usage remains unchanged, as ComPO does not store individual perturbation vectors $\\{y_i\\}_{1 \leq i \leq m}$ -- only the running average gradient estimates are maintained (see Line 5 of Algorithm 2).
>
> 3. **Only the output projection layer is tuned. It remains unclear whether full-model or larger-parameter variants would preserve gains or suffer instability.**
>
> We agree that exploring full-model or larger-parameter variants is important. We extended our experiments to perturb 3 layers (MLPs in layers 30–31 and the output layer) in Mistral-7B-Instruct, keeping all other parameters the same as in the paper. The AlpacaEval2 (AE) and ArenaHard (AH) results are:
>
> $$
> \\begin{array}{c|cc|c}
> \\textbf{Layer perturbed (parameters)} & \\textbf{AE-WR \\% - mean}\\,\\pm\\,\\textbf{std (best)} & \\textbf{AE-LC \\% - mean}\\,\\pm\\,\\textbf{std (best)}\\ & \\textbf{AH (GPT-4.1)-WR  \\% - mean}\\,\\pm\\,\\textbf{std (best)}\\\\ \\hline
> 1 (0.13B)  & 17.50\\,\\pm\\,0.65\\;(18.32) & 25.02\\,\\pm\\,0.91\\;(26.17) & 10.80\\,\\pm\\,0.21 \\;(11.0)\\\\
> 3 (0.25B) & 18.19\\,\\pm\\,0.81\\;(19.38)   & 26.00\\,\\pm\\,0.89\\;(27.09)  & 11.26\\,\\pm\\,0.36 \\;(11.7) \\\\
> \\end{array}
> $$
>
> Notably, we adopt the most recent version of Arena‑Hard with GPT‑4.1 as the judge, available only after we wrap up the initial submission. GPT-4.1 is a judge much stronger than GPT‑4 Turbo. The arrival of a new judge significantly impacts evaluation outcomes: for example, DPO’s WR drops from 14.4 (under GPT‑4 Turbo) to 10.5 (under GPT‑4.1), while ComPO's results show **consistent improvement** under this stronger and more robust judge.
>
> Perturbing more layers leads to better performance due to a richer gradient direction set. The peak GPU memory increases mildly from 16.3GB to 16.7GB, and running time per 600 perturbations takes 60 seconds (vs. 50 seconds), which we consider reasonable.
>
> To ensure stability, ComPO uses a gradient entry threshold $\lambda_g$, which selectively updates only high-magnitude gradient entries. We conducted ablation studies under $m=3300$, each with 5 independent trials:
>
> $$
> \\begin{array}{c|c|cc}
> \\textbf{Threshold $\\lambda_{g}$} & \\textbf{\\% updated gradient entry} & \\textbf{WR \\% – mean}\\,\\pm\\,\\textbf{std (best)} & \\textbf{LC \\% – mean}\\,\\pm\\,\\textbf{std (best)}\\\\
> \\hline
> 0       & 100\\%  & 15.72\\,\\pm\\,0.77\\;(16.34) & 23.42\\,\\pm\\,1.03\\;(24.28)\\\\
> 0.00004 &  63\\%  & 16.02\\,\\pm\\,0.69\\;(16.69) & 24.01\\,\\pm\\,0.91\\;(25.10)\\\\
> 0.00018 &   6\\%  & 19.02\\,\\pm\\,0.62\\;(20.15) & 26.06\\,\\pm\\,0.81\\;(27.27)\\\\
> 0.00022 &   1\\%  & 19.21\\,\\pm\\,0.58\\;(20.25) & 25.91\\,\\pm\\,0.95\\\;(27.14)\\\\
> 0.00025 &  0.15\\% & 16.10\\,\\pm\\,0.11\\;(16.21) & 23.82\\,\\pm\\,0.23\\;(24.00)\\\\
> \\end{array}
> $$
>
> We found that setting $\lambda_g$ to retain 1\%-5\% of entries offers the best trade-off between performance and stability. Too high or too low thresholds degrade performance.
>
> 4. **Could you clarify how sensitive ComPO’s performance is to the choice of threshold for separating noisy vs. clean preference pairs? Would a more adaptive or learned approach outperform a fixed threshold?**
>
> Thank you for raising this point. We evaluated ComPO's sensitivity to the threshold $\delta \in \\{1, 3, 5\\}$, which is used to distinguish clean vs. noisy preference pairs. Results show that performance remains largely consistent across these values (we use $\delta=3$ in the current paper). We also analyzed the number of successful perturbation oracles per noisy pair and found no clear trend across thresholds -- even small-margin pairs (e.g., $\delta=0.5$) often yield comparable feedback. This indicates that ComPO is robust to small‑to‑moderate changes in $\delta$. At extreme values (e.g., $\delta=50$), we observe a sharp increase in successful perturbations, potentially making such pairs suitable for DPO. This observation supports the idea that a more adaptive or learned threshold may outperform a fixed one. This requires a separate study and is a promising direction for future work.
>
> 5. **Your current scheme “hard-switches” objectives -- DPO on clean pairs, ComPO on noisy ones. Have you explored a soft interpolation, weighting DPO and ComPO by a confidence score derived from the reference model?**
>
> We agree that a soft interpolation between DPO and ComPO, based on a reference-model-derived confidence score, is a compelling idea. Intuitively, cleaner data should receive more weight on DPO, and noisier ones do more on ComPO. However, defining the "noise level" of a dataset remains nontrivial without relying on thresholds such as $\delta$. Even with such a measure, determining a suitable confidence score and interpolation scheme would require further study. While we believe this could lead to improved performance via a unified objective, we consider it beyond the scope of this work and leave it for future research.
>
> 6. **The organization of the paper could be improved for better clarity. In particular, Section 3 is titled “Main Results,” but it mixes algorithmic design, theoretical analysis, and practical implementation details.**
>
> Thank you for pointing this out. We agree that Section 3 currently conflates design, theory, and implementation aspects. In the revised version, we will restructure this section to clearly separate and delineate these components.
>
> 7. **The paper lacks a high-level pipeline diagram, does not define how Win-Rate (WR) and Length-Controlled Win-Rate (LC) are computed, and contains broken appendix references (e.g., line 272 cites Appendix B instead of E), all of which hinder clarity.**
>
> We appreciate your suggestions for improving clarity. In the revised manuscript, we will include a high-level pipeline diagram, clearly define Win Rate (WR) and Length-Controlled Win Rate (LC), and correct all broken or misreferenced appendix citations (e.g., line 272).
>
> We thank you again for your detailed reading and your constructive input! We hope and trust that our replies have alleviated your concerns, and we look forward to an open-minded discussion if any such concerns remain.

---

> > ### Comment · Reviewer_Hemu · 2025-08-05
> >
> > Many thanks for your detailed response. Most of my questions have been addressed. I will maintain my accept decision to this paper.

---

> ### Author Response · Authors · 2025-08-05
> **Thank you for your effort and time!**
>
> Dear Reviewer Hemu,
>
> We truly appreciate your time for such insightful and detailed review; the suggestions are very helpful for us to further strengthen ComPO.
>
> Thank you again for your support to our work!
>
> ---
>
> Sincerely,
>
> Authors of Paper 202

---

### Note · Authors · 2025-08-12

Dear Area Chair and All the Reviewers,

First, we sincerely appreciate your dedicated effort in coordinating the reviewers. All four reviewers actively engaged in the rebuttal process. This would not have been possible without you.

We also again extend our thanks to all the four reviewers for providing insightful feedback and for engaging in constructive discussions during the rebuttal.

ComPO is a novel, theoretically grounded method for LLM preference alignment that uses comparison oracles to turn noisy preference pairs -- often ineffectively utilized by DPO methods -- into a valuable training signal. Its key contributions include:

- Methodological novelty – reframes alignment as comparison-oracle optimization with convergence guarantees.

- Scalable and compatible – works as a plug-in to methods like DPO and SimPO, or works alone from their checkpoints.

- Practical impact – unlocks noisy preference data and shows robustness under stronger evaluators.

During discussions, all reviewers acknowledged the paper’s originality, soundness, and applicability. Multiple reviewers either raise or maintain positive scores. The consensus is that ComPO is a creative, well-supported advance with clear utility for the alignment community. We are encouraged by their recognition of ComPO as a novel and creative method, and we believe this work meaningfully bridges classic optimization theory with practical LLM applications.

Once again, we thank you and all four reviewers for your time and effort. We hope that ComPO will inspire further research at the intersection of optimization and LLMs.

---

Sincerely,

Authors of Paper 202

---

### Decision · Program_Chairs · 2025-09-17

**Decision:**

Accept (poster)

**Comment:**

The reviewers agree that RL with the comparison oracle proposed in this paper is novel and well-motivated. The theoretical grounding and the convergence proof further strengthen the methodological contribution. Ablations studies and experiments have demonstrated the robustness of the method.